
# Vorticity-Divergence semi-Lagrangian Global Atmospheric Model SL-AV20: Dynamical Core

Mikhail Tolstykh[1,2,3], Vladimir Shashkin[1], Rostislav Fadeev[1,3], and Gordey Goyman[3,1]

[1]Institute of Numerical Mathematics, Russian Academy of Sciences, Moscow, Russia
[2]Hydrometeorological Centre of Russia, Moscow, Russia
[3]Moscow Institute of Physics and Technology (State University), Moscow, Russia

*Correspondence to:* Vladimir Shashkin (vvshashkin@gmail.com)

**Abstract.** SL-AV (Semi-Lagranginan Absolute Vorticity) is a global atmospheric model. Its latest version SL-AV20 provides global operational medium-range weather forecast with 20km resolution over Russia. The lower resolution configurations of SL-AV20 are being tested for seasonal prediction and climate modeling.

The article presents the model dynamical core. Its main features are vorticity-divergence formulation at the unstaggered grid,
high-order finite-difference approximations, semi-Lagrangian semi-implicit discretization and the reduced latitude-longitude grid with variable resolution in latitude.

The accuracy of SL-AV20 numerical solutions using reduced lat-lon grid and the variable resolution in latitude is tested with two idealized testcases. The results agree well with other published model solutions. It is shown that the use of the reduced grid having up to 25% less grid points than the regular grid does not significantly affect the accuracy. Variable resolution in
latitude allows to improve the accuracy of solution in the region of interest.

## 1 Introduction

Atmospheric General Circulation Models (AGCM) are basic tools for weather forecasting from several days to seasons. Also, such models are essential building blocks of Earth system models used for climate simulations. AGCM consists of dynamical core responsible for solution of dynamics equations of resolvable flow component and the package representing sub-grid scale
processes (often in parameterized way).

SL-AV20 is the latest version of hydrostatic AGCM developed at the Institute of Numerical Mathematics, Russian Academy of Sciences (INM RAS) in cooperation with the Hydrometeorological centre of Russia (HMCR). SL-AV is the model acronym (semi-Lagrangian, based on Absolute-Vorticity equation), 20 indicates the targeted horizontal resolution over the territory of Russia. The SL-AV20 dynamical core is developed by the coauthors, while the most part of subgrid-scale parameterizations is
adopted from ALADIN/LACE model (Geleyn et al., 1994; Gerard et al., 2009).

SL-AV20 is accepted as a basic method for operational medium-range weather forecast in HMCR in 2015. The lower resolution version of this model will likely be applied in HMCR long-range forecasting system. Also, SL-AV20 is considered as a starting point for developing the new atmospheric component of INM climate model (INMCM, Volodin et al., 2010). Among the papers that discuss SL-AV model numerics e.g. Tolstykh (2002), Tolstykh (2003), Tolstykh and Shashkin (2012),



Shashkin and Tolstykh (2014), there is no one that would put all the developments together. This article describes the present state of the SL-AV20 dynamical core.

While developing a dynamical core, we want it to have a given accuracy with respect to test solutions combined with a given wall-clock time and minimum number of processors necessary to calculate these solutions. We call this computational

efficiency. Furthermore, as the dynamical core can be used at a range of horizontal and vertical resolutions typical for numerical weather prediction and climate simulations, it is desirable to maintain dynamical core computational efficiency at maximum possible range of resolutions. This is somewhat contradictory requirement. Indeed, using a global dynamical core at maximum possible resolution of first kilometers ultimately requires dynamical core to use efficiently up to tens of thousand cores. This is not so easy to achieve if a semi-implicit time integration scheme is used.

Given typical applications of SL-AV20 model mentioned above but also limited computer resources available, our approach to the dynamical core design is based on the following choices. We use semi-implicit (SI) time stepping scheme (Robert et al., 1985) and semi-Lagrangian (SL) treatment of advection (a review –  Staniforth and Côté, 1991). This combination allows to circumvent Courant-Friedrichs-Lewy stability limitation for both wind speed and inertia-gravity wave phase speed. Practically, this means that the time-step is much larger than with the Eulerian treatment of advection and/or explicit time-

stepping scheme, however, at the cost of solving Helmholtz equation at each time-step and having larger communication pattern in parallel implementation. The unstaggered grid is used, i.e. scalar and vector variables are stored at the same grid points. Therefore, only one set of upstream trajectories needs to be computed for SL advection scheme (C-grid requires 3 sets). Also, this improves trajectory computation accuracy, since the velocity components are stored at arrival points and no spatial interpolation or averaging is needed to compute the displacement terms at arrival point.

Following the results of Randall (1994), the vertical component of relative vorticity and the horizontal divergence are used as prognostic variables to achieve good inertia-gravity and Rossby wave dispersion properties at the unstaggered grid. However, the reconstruction of wind velocity from vorticity and divergence is needed at each time-step. We use the direct inversion of relative vorticity and divergence finite-difference formulae avoiding solution of Poisson problems for stream function and velocity potential. This leads to the accurate and efficient solver Tolstykh and Shashkin (2012).

Fourth-order finite-difference formulae are used to compute gradient, divergence, and vorticity operators. Compact finite-differences were used in early versions of the SL-AV model (Tolstykh, 2002) for their smaller truncation errors. To improve parallel efficiency, compact finite-differences are abandoned now except Helmholtz and velocity reconstruction solvers.

The important question for the global atmospheric model is the choice of the horizontal grid on the sphere. The regular latitude-longitude grid that served in the atmospheric modelling for decades is now ending its era. The grid-points clustering

near the poles forces to use either very small time-step for stability or special numerical techniques (polar filters, SL advection) that needs excessive data transfer in the polar regions and leads to scalability problems. Moreover, physical grid-spacing in latitude in the polar regions can be order of magnitude (or even more) larger than the grid-spacing in longitude. This presents a difficulty for parametrizations of sub-grid scale processes. A significant effort is made to develop models based on icosahedral (e.g. Zangl et al., 2015), Voronoy/Delaunay (e.g. Skamarock et al., 2012), cubed-sphere (e.g. Fournier et al., 2004), Yin-Yang

(Quaddouri and Lee, 2011) or some less popular grids on the sphere.





Unfortunately, all quasi-uniformly spaced grids proposed to date suffer from one or more of the following problems (as discussed in Staniforth and Thuburn, 2012): disbalance between vector and scalar degrees of freedom (grids with triangular or hexagonal/pentagonal cells), non-orthogonality of underlaying coordinate system (cubed-sphere), overset regions or grid transition (Yin-Yang, cubed sphere). These issues can degrade the accuracy of atmospheric circulation simulation, cause grid-imprinting and/or occurrence of unphysical wave modes and some other problems.

SL-AV20 is formulated using the reduced latitude-longitude grid suggested in Kurihara (1965) but with different design. We believe that using reduced grid can alleviate most of the polar problems of the regular latitude-longitude grid. Moreover, the reduced latitude-longitude grid is free of problems specific to more complex quasi-uniformly spaced grids, and is relatively easy to implement. As shown in Tolstykh and Shashkin (2012), the accuracy problems of reduced grid computations reported by Williamson (2007), Staniforth and Thuburn (2012) can be overcome with proper construction of this grid (Fadeev, 2013) and using high-order discretizations.

Another feature of SL-AV20 is the possibility to use variable resolution in latitude. This approach is especially suitable for Russia territory which is stretched for almost $180°$ in longitude. Variable resolution in latitude with the ability to use non equatorially symmetric grid reduction allows us to refine resolution in the region of interest (midlatitudes of Northern hemisphere) and to coarsen it in other regions (e.g. Southern hemisphere).

The article is organized as follows. The governing equations are listed in Sect. 2, Sect. 3 briefly presents SL-AV20 semi-Lagrangian advection scheme. The details of semi-implicit temporal discretization are given in Sect. 4, the numerical techniques for horizontal and vertical discretization are presented in Sect. 5. The solvers for Helmholtz problem and wind velocity reconstruction are presented in Sect. 6, Sect. 7 discusses the dissipation mechanisms used in the model. Section 8 describes SL-AV20 parallel implementation. Numerical experiments are described in Sect. 9.

## 2 Governing equations

Governing equations of the SL-AV model are derived from the set of hydrostatic shallow-atmosphere primitive equations on the sphere (Holton, 2004, chapter 2). The hybrid coordinate $\eta$ of Simmons and Burridge (1981) is used in the vertical.

The spherical longitude and latitude are $(\lambda, \varphi)$, $\boldsymbol{r}$ is the vector pointing from the center of the sphere to the point on its surface, $a$ is the radius of the sphere (notations used in the article are summarized in Appendix A). Standard definitions of horizontal $\nabla$ operator and Lagrangian derivative $\frac{\mathrm{d}}{\mathrm{d}t}$ are used . The pressure is $p(\eta) = A(\eta)p_0 + B(\eta)p_\mathrm{s}$, $p_0$ is a constant reference pressure and $p_\mathrm{s}$ is the surface pressure.

The horizontal wind velocity is $\boldsymbol{V}$ with $u$ and $v$ being its zonal and meridional components respectively. $D = \nabla \cdot \boldsymbol{V}$ and $\zeta = \boldsymbol{k} \cdot \nabla \times \boldsymbol{V}$ are the horizontal divergence and vertical component of relative vorticity, $\boldsymbol{k} = \boldsymbol{r}/a$ is the vertical unit vector. The vertical velocity in the hybrid coordinate system is $\dot{\eta}$. The Coriolis parameter $f = 2|\boldsymbol{\Omega}|\sin\varphi$, $\boldsymbol{\Omega}$ is the Earth's angular velocity. $\Phi$ is the geopotential, $T$ is the temperature, $\Phi_\mathrm{s}$ is the surface geopotential. We also use virtual temperature $T_v = \frac{R_\mathrm{moist}}{R_d}T$; $R_d$ is the dry air gas constant and $R_\mathrm{moist} = (1 - q - \sum q_i)R_d + R_v q$, where $R_v$ is the gas constant of water vapor, $q$ is water vapor specific concentration, $q_i$ are liquid and solid water species specific concentrations. The heat capacity of moist and dry air at





constant pressure are $c_p$ and $c_{pd}$, $c_p$ includes contributions from all water species presented. The sources/sinks of arbitrary quantity $\psi$ due to subgrid/diabatic processes are denoted by $F_\psi$.

We begin with the equation for the vertical component of absolute vorticity $\zeta$ obtained by applying $\boldsymbol{k} \cdot \nabla \times$ operator to the momentum equations:

$$\frac{\mathrm{d}}{\mathrm{d}t}\left(\zeta + f\right) = -\left(\zeta + f\right)D - J_\zeta + F_\zeta \tag{1}$$

$$J_\zeta = \frac{B(\eta)p_{\mathrm{s}}}{A(\eta)p_0 + B(\eta)p_{\mathrm{s}}}\frac{R_d}{a^2\cos\varphi}\left(\frac{\partial T_v}{\partial\lambda}\frac{\partial\ln p_{\mathrm{s}}}{\partial\varphi} - \frac{\partial T_v}{\partial\varphi}\frac{\partial\ln p_{\mathrm{s}}}{\partial\lambda}\right) + \frac{1}{a\cos\varphi}\left(\frac{\partial\dot\eta}{\partial\lambda}\frac{\partial v}{\partial\eta} - \cos\varphi\frac{\partial\dot\eta}{\partial\varphi}\frac{\partial u}{\partial\eta}\right), \tag{2}$$

Equation for the horizontal divergence $D$ is derived from the discrete form of momentum equations (see Sect. 4 for details). The momentum equations are written in the vector form of Bates et al. (1993) using advective Coriolis term (Rochas, 1990):

$$\left(\frac{\mathrm{d}\boldsymbol{V}}{\mathrm{d}t} + 2\boldsymbol{\Omega}\times\frac{\mathrm{d}\boldsymbol{r}}{\mathrm{d}t}\right)_{\mathrm{H}} = -\nabla\Phi - \frac{B(\eta)p_{\mathrm{s}}}{A(\eta)p_0 + B(\eta)p_{\mathrm{s}}}R_d T_v \nabla\ln p_{\mathrm{s}} + \boldsymbol{F_V}, \tag{3}$$

subscript $H$ denotes projection on the surface of the sphere.

Also, there is an option to use divergence equation obtained in analytic way applying $\nabla\cdot$ operator to momentum equations (Eq. (3) can be used, however the derivation is simpler with the component form Holton, 2004):

$$\frac{\mathrm{d}D}{\mathrm{d}t} + D^2 = -\nabla^2\Phi - \frac{B(\eta)p_{\mathrm{s}}}{A(\eta)p_0 + B(\eta)p_{\mathrm{s}}}\nabla\cdot(R_d T_v \nabla\ln p_{\mathrm{s}}) + \zeta f - \frac{u}{a}\frac{\partial f}{\partial\varphi} + J_D + F_D, \tag{4}$$

$$J_D = -\frac{1}{a^2\cos\varphi}\left\{\frac{\partial\big((u^2 + v^2)\sin\varphi\big)}{\partial\varphi} + 2\left[\frac{\partial}{\partial\varphi}\left(u\frac{\partial v}{\partial\lambda}\right) - \frac{\partial}{\partial\lambda}\left(u\frac{\partial v}{\partial\varphi}\right)\right] + \frac{1}{a\cos\varphi}\left(\frac{\partial\dot\eta}{\partial\lambda}\frac{\partial u}{\partial\eta} + \cos\varphi\frac{\partial\dot\eta}{\partial\varphi}\frac{\partial v}{\partial\eta}\right)\right\}. \tag{5}$$

Using analytic divergence equation (4) gives a possibility to use the same spatial approximation for $\nabla^2\Phi$ in both explicit and implicit parts of time-discrete equations. Therefore, we can expect better dispersion properties for the shortest inertia-gravity waves (according to Caluwaerts et al., 2015), however, at the cost of computing additional non-linear terms. It is interesting to note that unlike Heikes and Randall (1995) who formulated vorticity and divergence equations in terms of true scalars ($\zeta$, $D$, $\Phi$, stream function and velocity potential) we use components of vector quantities $\boldsymbol{V}$ and $\nabla\Phi$.

Thermodynamic equation is readily formulated for virtual temperature $T_v$. The energy conversion term $\frac{1}{p}\frac{\mathrm{d}p}{\mathrm{d}t}$ is rewritten in terms of $\ln p_{\mathrm{s}}$ and analogue of vertical velocity $\dot s = \frac{1}{p_{\mathrm{s}}}\frac{\partial p}{\partial\eta}\dot\eta$ (McDonald and Haugen, 1993):

$$\frac{\mathrm{d}\big(T_v + \gamma(\eta)\Phi_{\mathrm{s}}\big)}{\mathrm{d}t} - \frac{R_{\mathrm{moist}}T_v}{c_p}\left(\frac{p_{\mathrm{s}}}{A(\eta)p_0 + B(\eta)p_{\mathrm{s}}}\dot s + \frac{B(\eta)p_{\mathrm{s}}}{A(\eta)p_0 + B(\eta)p_{\mathrm{s}}}\frac{\mathrm{d_H}\ln p_{\mathrm{s}}}{\mathrm{d}t}\right) = F_{T_v} + \gamma(\eta)\boldsymbol{V}\cdot\nabla\Phi_{\mathrm{s}} + \dot\eta\frac{\partial\gamma(\eta)}{\partial\eta}\Phi_{\mathrm{s}}, \tag{6}$$

$$F_{T_v} = \frac{R_{\mathrm{moist}}}{R_d}F_T + T\left[\left(\frac{R_{\mathrm{v}}}{R_d} - 1\right)F_q - \sum F_{q_i}\right], \tag{7}$$

where $\frac{\mathrm{d_H}}{\mathrm{d}t}$ is horizontal Lagrangian derivative with neglected vertical displacement. The term $\gamma(\eta)\Phi_{\mathrm{s}}$ was proposed by Ritchie and Tanguay (1996) to suppress spurious orographic resonance, also, it smooths temperature field near orographic variations and contribute positively to the accuracy of temperature advection.





The continuity equation in the form of mass conservation for an arbitrary Lagrangian cell $\mathcal{V}(t)$ can be written as

$$\frac{1}{g}\frac{\mathrm{d}}{\mathrm{d}t}\int\limits_{\mathcal{V}(t)}\left(\frac{\partial p}{\partial \eta}\right)d\mathcal{V} = 0, \tag{8}$$

is rewritten in terms of $\ln p_s$ and $\dot{s}$ similar to McDonald and Haugen (1993):

$$\frac{\partial B}{\partial \eta}\frac{\mathrm{d_H}}{\mathrm{d}t}\left(\ln p_s + \frac{\Phi_s}{R_d T_{\mathrm{const}}}\right) = -\frac{\frac{\partial A}{\partial \eta}p_0 + \frac{\partial B}{\partial \eta}p_s}{p_s}D - \frac{\partial \dot{s}}{\partial \eta} + \frac{\partial B}{\partial \eta}\boldsymbol{V}\cdot\nabla\left(\frac{\Phi_s}{R_d T_{\mathrm{const}}}\right), \tag{9}$$

the term $\left(\frac{\Phi_s}{R_d T_{\mathrm{const}}}\right)$ was also proposed by Ritchie and Tanguay (1996).

The hydrostatic balance equation is

$$\frac{\partial \Phi}{\partial \eta} = -R_d T_v \frac{\partial \ln p}{\partial \eta}. \tag{10}$$

The equations for transport of water vapour and other water species are all written in the same form:

$$\frac{\mathrm{d}q}{\mathrm{d}t} = F_q, \tag{11}$$

$$\frac{\mathrm{d}}{\mathrm{d}t}\int\limits_{\mathcal{V}(t)}\left(\frac{\partial p}{\partial \eta}\right)q\mathrm{d}\mathcal{V} = \int\limits_{\mathcal{V}(t)}\left(\frac{\partial p}{\partial \eta}\right)F_q\mathrm{d}\mathcal{V}, \tag{12}$$

the finite-volume form (12) is used for locally mass-conserving discretization.

Boundary conditions for the above Eqs. (1)–(12) are $\dot{\eta} = 0$ at the lower $\eta = 1$ and upper $\eta = \eta_{top}$ boundaries. Also, it is assumed that $B(\eta) = 1$, $A(\eta) = 0$ when $\eta = 1$, normally $B = 0$ above some $\eta_p > \eta_{top}$, however model can work in the pure $\sigma$-coordinate mode, when $B(\eta) = \eta$, $A(\eta) = 0$ everywhere.

# 3   Semi-Lagrangian advection

## 3.1   Conventional semi-Lagrangian advection

Lagrangian time derivatives in the prognostic equations given in Sect. 2 are approximated in time as $\mathrm{d}\psi/\mathrm{d}t = (\psi^{n+1} - \psi_*^n)/\Delta t$, where superscript indicates time level $t^n = n\Delta t$, subscript $*$ indicates that quantity $\psi$ is evaluated at the departure position of Lagrangian parcel (at $t^n$). Under the SL approach, each point of fixed computational grid is the arrival position of some

Lagrangian particle. The advection equation $\mathrm{d}\psi/\mathrm{d}t = 0$ is discretized in time as $\psi^{n+1} = \psi_*^n$. SL approximation is not subjected to the Courant-Friedrichs-Lewy stability limitation, so $\Delta t$ can be chosen from accuracy considerations that greatly contributes to the computational efficiency.

The departure position of the Lagrangian parcel arriving to some grid point can be approximately found by integrating the equation $\frac{\mathrm{d}\boldsymbol{r}}{\mathrm{d}t} = \boldsymbol{V}$ one step back in time. Following SETTLS scheme (Hortal, 2002), we approximate this equation as

$$\frac{\boldsymbol{r}^{n+1} - \boldsymbol{r}_*^n}{\Delta t} = \frac{1}{2}\left(\boldsymbol{V}^n + \boldsymbol{V}_*^{(n+1)e}\right), \tag{13}$$





where $\boldsymbol{V}^{(n+1)e} = 2\boldsymbol{V}^n - \boldsymbol{V}^{n-1}$. Iterative approach is used to solve Eq. (13):

$$\boldsymbol{r}^n_{*m+1} = \boldsymbol{r}^{n+1} - \frac{\Delta t}{2}\Big(\boldsymbol{V}^n + \boldsymbol{V}^{(n+1)e}_{*m}\Big), \tag{14}$$

where $*m$ indicates the departure point position found at $m$-th iteration. Additional geometric approximations are needed iterating Eq. (14) to account for spherical geometry and shallow atmosphere approximation (see Temperton et al., 2001).

The quantities at the departure positions of Lagrangian particles are evaluated using interpolation. We use 3D tensor-product cubic Hermite interpolation for advective terms of equations (the terms that arise from SL approximation of $\mathrm{d}/\mathrm{d}t$). Trilinear interpolation is used for components of $\boldsymbol{V}$ in iterative process (14) and non-advective terms of equations. The change in the direction of coordinate system unit vectors between departure and arrival positions should be taken into account when interpolating the vector components:

$$\frac{\mathrm{d}\boldsymbol{\psi}}{\mathrm{d}t} = \frac{1}{\Delta t}\left(\begin{pmatrix} \psi_\lambda \\ \psi_\varphi \end{pmatrix} - \mathfrak{R}\begin{pmatrix} \psi_{\lambda*} \\ \psi_{\varphi*} \end{pmatrix}\right), \tag{15}$$

where $\mathfrak{R}$ is the rotation matrix (see Temperton et al., 2001, detailed derivation in Staniforth et al., 2010).

### 3.2    Mass-conservative semi-Lagrangian advection

The disadvantage of semi-Lagrangian approach as formulated in Sect. 3.1 is the lack of tracer mass conservation. To address this problem, SL-AV20 incorporates finite-volume SL Conservative Cascade Scheme (CCS-3D, Shashkin et al., 2016). Finite-

volume SL discretization starts with the advection equation in form (12) where the arrival volume $\mathcal{V}(t^{n+1})$ coincides with grid cell $\mathcal{V}_{ijl}$ ($l$ is vertical index):

$$\left(\frac{\partial p}{\partial \eta}q\right)^{n+1}_{\mathcal{V}_{ijl}} S_{ij}\Delta\eta_k = \int\limits_{\mathcal{V}^*_{ijl}}\left(\frac{\partial p}{\partial \eta}\Big(q + F_q\Delta t\Big)\right)^n d\mathcal{V}. \tag{16}$$

$(.)_{\mathcal{V}_{ijl}}$ is quantity averaged over $\mathcal{V}_{ijl}$, $S_{ij}$ and $S_{ij}\Delta\eta_l$ are the cell horizontal square and volume respectively. Cell-averaged tracer density $\frac{\partial p}{\partial \eta}q$ is chosen as prognostic variable.

The departure cell $\mathcal{V}(t^n) = \mathcal{V}^*_{ijl}$ is defined by its vertices i.e. the departure positions of Lagrangian parcels arriving to the vertices of $\mathcal{V}_{ijl}$ at $t^{n+1}$. Departure cells vertices are found with interpolation of known coordinates of departure cells centers. The latter are departure positions of the Lagrangian parcels arriving to the grid points computed using Eq. (13).

     Approximation of integrals over the departure cells is the main problem of FVSL discretization. Geometrical approximations of departure cells shape and reconstruction of $(q\partial p/\partial \eta)^n$ subgrid distribution are employed (Shashkin et al., 2016) to solve

this problem efficiently.



## 4   Time discretization

### 4.1   Basic semi-implicit formulation

The non-advective terms of prognostic equations (from Sect. 2) are approximated using the combination of Crank-Nicolson scheme with the pseudo-second order decentering (Temperton et al., 2001) for linear terms and SETTLS scheme (Hortal, 2002) for non-linear terms. For the equation

$$\frac{\mathrm{d}\psi}{\mathrm{d}t} = L\psi + N(\psi), \tag{17}$$

where $\psi$ is an arbitrary scalar or vector variable the scheme writes as

$$\frac{\psi^{n+1} - \psi_*^n}{\Delta t} = \frac{1}{2}\left(N(\psi)_*^{(n+1)e} + N(\psi)^n\right) + \frac{1+\epsilon}{2}L\psi^{n+1} + \frac{1+\epsilon}{2}L\psi_*^n - \frac{\epsilon}{2}\left(L\psi_*^{(n+1)e} + L\psi^n\right). \tag{18}$$

Absolute vorticity Eq. (1) is discretized in time as follows:

$$\zeta^{n+1} + \frac{\Delta t}{2}fD^{n+1} = R_\zeta, \tag{19}$$

where $R_\zeta$ is combination of time-level $n$ and extrapolated in time $((n+1)_e)$ quantities (see Appendix B for details on right-hand-side terms of SI system of equation), no decentering is applied to term $fD$. We use the background state with constant reference temperature $\bar{T}$ and reference pressure $\bar{p} = A(\eta)p_0 + B(\eta)\bar{p}_s$ ($\bar{p}_s$ is a constant) to extract the linear terms in Eqs. (3), (4), (6), (9). First, the pressure gradient term of momentum equations (3) is split into linear and non-linear contribution:

$$\nabla\left(\Phi_s - \int_1^\eta R_d T_v \mathrm{d}\ln p\right) + \frac{B(\eta)p_s}{A(\eta)p_0 + B(\eta)p_s}R_d T_v \nabla\ln p_s =$$

$$\underbrace{\left(\nabla G\right)}_{\text{linear term}} \underbrace{-\nabla\int_1^\eta R_d\left(T_v - \bar{T}\right)\mathrm{d}\ln\frac{p}{\bar{p}} + \frac{B(\eta)p_s}{A(\eta)p_0 + B(\eta)p_s}R_d\left(T_v - \bar{T}\right)\nabla\ln p_s}_{\text{non-linear term}}, \tag{20}$$

$$G = \Phi_s - \int_1^\eta R_d\left(T_v - \bar{T}\right)\mathrm{d}\ln\bar{p} + R_d\bar{T}\ln p_s, \tag{21}$$

here we use the fact that $\nabla\int_1^\eta R_d\bar{T}\mathrm{d}\ln p = R_d\bar{T}[\nabla p(\eta)/p(\eta) - \nabla\ln p_s] = R_d\bar{T}[B(\eta)p_s/(A(\eta)p_0 + B(\eta)p_s) - 1]\nabla\ln p_s$. Similar relation is derived for $\nabla^2$ of pressure in the analytic divergence Eq. (4). The approximation of vertical integrals is described in Sect. 5.5.

The time-discretized divergence equation is written as

$$D^{n+1} + \frac{(1+\epsilon)}{2}\Delta t\nabla^2 G^{n+1} = \nabla \cdot \boldsymbol{R_V}, \tag{22a}$$

$$D^{n+1} - \frac{\Delta t}{2}f\zeta^{n+1} + \frac{(1+\epsilon)}{2}\Delta t\nabla^2 G^{n+1} = R_D, \tag{22b}$$




Eq. (22a) is the divergence equation derived from momentum equation (3) discretized in time via Eq. (18), $\boldsymbol{R_V}$ is the right-hand-side (RHS) of corresponding vector equation. The RHS term $\nabla \cdot \boldsymbol{R_V}$ is calculated with the discrete divergence operator from Sect. 5.2. Equation (22b) is based on analytic divergence Eq. (4). This equation allows to use the same Crank-Nicolson discretization in time for $-f\zeta$ and $\nabla^2 G$, the two terms representing geostrophic balance.

In the thermodynamic equation (6), the Lagrangian time derivative of $\ln p_s$ is discretized in the SL manner in the linear part of energy conversion term, whereas it is replaced by the RHS of continuity Eq. (9) in the non-linear part. The resulting time discrete equation is

$$T_v^{n+1} - \frac{R_d \bar{T}}{c_{pd}} \frac{\bar{p}_s}{A(\eta)p_0 + B(\eta)\bar{p}_s} \left( B(\eta) \ln p_s^{n+1} + \frac{1+\epsilon}{2} \Delta t \dot{s}^{n+1} \right) = R_T. \tag{23}$$

The continuity equation (9) is discretized as follows:

$$\frac{\partial B}{\partial \eta} \ln p_s^{n+1} + \frac{1+\epsilon}{2} \Delta t \frac{(\partial A/\partial \eta)p_0 + (\partial B/\partial \eta)\bar{p}_s}{\bar{p}_s} D^{n+1} + \frac{1+\epsilon}{2} \Delta t \frac{\partial \dot{s}^{n+1}}{\partial \eta} = R_P. \tag{24}$$

Integration of Eq. (24) from model top to the ground provides the expression for $\ln p_s^{n+1}$ independent of $\dot{s}^{n+1}$, whereas the integration to the arbitrary level $\eta$ gives the $\dot{s}^{n+1}$:

$$\left(1 - B(\eta_{top})\right) \ln p_s^{n+1} = -\frac{1+\epsilon}{2} \Delta t \int\limits_{\eta=\eta_{top}}^{\eta=1} \left( \frac{(\partial A/\partial \eta)p_0 + (\partial B/\partial \eta)\bar{p}_s}{\bar{p}_s} D^{n+1} + R_P \right) d\eta, \tag{25}$$

$$\frac{1+\epsilon}{2} \Delta t \dot{s}^{n+1}(\eta) = -\left(B(\eta) - B(\eta_{top})\right) \ln p_s^{n+1} - \frac{1+\epsilon}{2} \Delta t \int\limits_{\eta=\eta_{top}}^{\eta=\eta} \left( \frac{(\partial A/\partial \eta)p_0 + (\partial B/\partial \eta)\bar{p}_s}{\bar{p}_s} D^{n+1} + R_P \right) d\eta, \tag{26}$$

the RHS of Eq. (25) can be easily substituted for $\ln p_s^{n+1}$ term in Eq. (26) providing equation for $\dot{s}^{n+1}$ which depends only on $D^{n+1}$ unknown quantity.

Equations (19), (22), (23), (25), (26) and the definition of linear geopotential $G$ (21) forms the linear system for time-level $n+1$ variables $(\zeta, D, T_v, \ln p_s, \dot{s}, G)$. The solution procedure for this system is as follows. First, if analytically-derived divergence equation (22b) is used, the relative vorticity $\zeta^{n+1}$ is excluded by substitution from Eq. (19). Then, Eqs. (25), (26) are substituted in (23) to obtain expression for $T_v^{n+1}$ that depends only on $D^{n+1}$. This expression and Eq. (25) are used to eliminate $T_v^{n+1}$ and $\ln p_s^{n+1}$ in the $G^{n+1}$ definition Eq. (21). Therefore, we obtain the pair of equations that contains only $G^{n+1}$ and $D^{n+1}$ unknowns. With the aid of vertical discretization (see Sect. 5.5) this pair can be written as

$$\boldsymbol{G} + \frac{1+\epsilon}{2} \Delta t \mathbf{M} \boldsymbol{D} = \boldsymbol{H}, \tag{27}$$

$$(1 + \alpha \frac{f^2 \Delta t^2}{4}) \boldsymbol{D} + \frac{1+\epsilon}{2} \Delta t \nabla^2 \boldsymbol{G} = \boldsymbol{R_D} + \alpha \frac{f \Delta t}{2} \boldsymbol{R_\zeta}, \tag{28}$$

where $\boldsymbol{G}$, $\boldsymbol{D}$ and $\boldsymbol{R_\psi}$ are columns of Nlev (number of model levels) components, with $l$-th component representing the corresponding horizontal field at the level $l$, $\boldsymbol{H} = \boldsymbol{\Phi}_s + R_d \mathbf{A} \boldsymbol{R_T} + \mathbf{M}' \boldsymbol{R_P}$, $\mathbf{M}$, $\mathbf{M}'$, $\mathbf{A}$ are the matrices of discrete vertical operators (resulting from vertical integration, see details in Appendix C). The coefficient $\alpha = 0$ if the standard divergence



equation (22a) is used and $\alpha = 1$ for analytically-derived divergence equation (22b), $\boldsymbol{R}_D$ corresponds to the RHS term of the relevant divergence equation.

Substituting $\boldsymbol{D}$ from Eq. (28) in Eq. (27) and using eigenvalue decomposition $\mathbf{M} = \mathbf{P}\boldsymbol{\Lambda}\mathbf{P}^{-1}$ results in Nlev 2D Helmholtz problems for components of $\mathbf{P}^{-1}\boldsymbol{G}$ vector. The solution procedure for Helmholtz problems is described in Sect. 6.2. Once $\boldsymbol{G}$

is found, we can find the divergence $\boldsymbol{D}$ with Eq. (27) and use it to calculate $\ln p_\mathrm{s}^{n+1}$, $\dot{s}^{n+1}$, $T_v^{n+1}$ and $\zeta^{n+1}$ (see Eqs., (25), (26), (23), (19)). The $n + 1$-th time-step calculations are finalized by reconstruction of the horizontal wind $\boldsymbol{V}^{n+1}$ as described in Sect. 6.1 and calculation of $\dot{\eta}$ and recalculation of $\dot{s}$ (Sect. 5.5).

We have also implemented an iterative time-integration scheme (Goyman, 2015) in addition to the time discretization described above. The basic discretization is used at first iteration to obtain first-guess $n + 1$ time-step fields $(\zeta, D, T, \ln p_\mathrm{s}, \boldsymbol{V})$. At

the second iteration, we replace the time-extrapolated non-linear terms $N^{(n+1)e}$ of the equations and time-extrapolated wind velocity $(u, v, \dot{\eta})^{(n+1)e}$ involved in the upstream trajectory computations with those obtained using $n + 1$ time-step first-guess values. Using this iterative approach allows to improve stability and use larger time-steps without degrading accuracy.

## 4.2 Inherently mass-conserving model semi-implicit formulation

Inherently mass-conserving (IMC) version of the model (Shashkin and Tolstykh, 2014) uses continuity equation in the finite-

volume form (8). The non-linearity of this equation is hidden into the trajectory computation or equivalently in the evolution of the Lagrangian cell $\mathcal{V}(t)$. The equation is linearized using orography dependent reference pressure profile $p^{ref} = Ap_0 + Bp_\mathrm{s}^{ref}$, where $p_\mathrm{s}^{ref} = 1013.25\mathrm{hPa} \times \exp(-\Phi_\mathrm{s}/(R_d \bar{T}))$. The following equation is

$$\frac{\mathrm{d}}{\mathrm{dt}} \int_{\mathcal{V}(t)} \left(\frac{\partial p'}{\partial \eta}\right) d\mathcal{V} = - \int_{\mathcal{V}(t)} \left(\nabla \cdot \left(\frac{\partial p^{ref}}{\partial \eta}\boldsymbol{V}\right) + \frac{\partial}{\partial \eta}\left(\dot{\eta}\frac{\partial p^{ref}}{\partial \eta}\right)\right) d\mathcal{V}, \tag{29}$$

where $p' = p - p^{ref}$. The equation for mass-conservative update of $p_\mathrm{s}$ is obtained with discretization of Eq. (29) using the

scheme (18) and its integration from model top to bottom:

$$\left(1 - B(\eta_{top})\right) S_{ij}\left(p_\mathrm{s}^{n+1} - p_\mathrm{s}^{ref}\right)_{S_{ij}} = \sum_{l=1}^{l=\mathrm{Nlev}} \left(-\frac{1+\epsilon}{2}\Delta t\nabla\cdot\left(\frac{\partial p^{ref}}{\partial \eta}\boldsymbol{V^{n+1}}\right) + \frac{\epsilon}{2}\Delta t\nabla\cdot\left(\frac{\partial p^{ref}}{\partial \eta}\boldsymbol{V^n}\right)\right)_{\mathcal{V}_{ijl}} S_{ij}\Delta\eta_l +$$

$$\sum_{l=1}^{l=\mathrm{Nlev}} \int_{\mathcal{V}_{ijl}^*} \left(\left(\frac{\partial p'}{\partial \eta}\right) - \frac{1+\epsilon}{2}\Delta t\nabla\cdot\left(\frac{\partial p^{ref}}{\partial \eta}\boldsymbol{V^n}\right) + \frac{\epsilon}{2}\Delta t\nabla\cdot\left(\frac{\partial p^{ref}}{\partial \eta}\boldsymbol{V^{(n+1)e}}\right)\right) d\mathcal{V}, \tag{30}$$

where $(p_\mathrm{s})_{S_{ij}}$ is the surface pressure averaged over $S_{ij}$. The departure volume integrals are evaluated using CCS-3D (Shashkin

et al., 2016).

Using orography dependent reference pressure profile is crucial for stability in mountainous regions. The term $\nabla\cdot\left(\frac{\partial p^{ref}}{\partial \eta}\boldsymbol{V^{n+1}}\right)$, however, cannot be expanded in terms of model prognostic variables, so it is very difficult to solve the system of equations similar to those formulated in Sect. 4.1, but with mass-conservative equation for $p_\mathrm{s}^{n+1}$ (30) instead of Eq. (25). Therefore, the following procedure is implemented. First, we run a time-step of non mass-conserving model version, solve the semi-implicit





system as it formulated in Sect. 4.1 and obtain $n+1$ time-step horizontal wind $\boldsymbol{V}^{n+1}$. Then the mass-conservative update of $p_\mathrm{s}$, Eq. (30) is calculated.

We achieve inherent mass-conservation, however, at a cost of introducing some inconsistency between the surface pressure and horizontal wind fields. This results in a slight noise in the regions with steep orography, but does not significantly affect the model accuracy. The greater problem not solved yet is the consistency between dry air and tracer mass so IMC model version now is rather a research option.

## 5 Spatial discretization

### 5.1 Horizontal grid

SL-AV20 model incorporates reduced latitude-longitude grid with variable resolution in latitude. The grid consists of points located on the latitudes $\varphi = \varphi_j$, $j \in [0, \mathrm{Nlat}]$ with longitude spacing $\Delta\lambda_j$, $(\lambda, \varphi)_{ij} = (i\Delta\lambda_j, \varphi_j)$. The number of points on a grid latitude is reduced polewards of equator. The pole points are assumed to be the grid latitudes. The regular latitude-longitude grid is the special case of the grid described above with constant latitude spacing and the same number of points on each grid latitude.

Formally, the model algorithms can work with any set of $\varphi_j$ and $\Delta\lambda_j$. However, the choice of these crucially affects accuracy. The grids are constructed with the algorithm of Fadeev (2013). This algorithm generates the set of $\varphi_j$ that satisfy grid smoothness constraints (to avoid spurious wave reflection at the regions of abrupt spacing change) and match the desired grid spacing latitude dependence as close as possible. Then, given $\varphi_j$ and the total number of grid points, the algorithm optimizes $\Delta\lambda_j$ to minimize the error of interpolation of an analytic test function to some finer grid.

### 5.2 Discretization of horizontal gradient, divergence and vorticity operators

The following 4-th order accurate formula is applied to evaluate first derivative:

$$\left(\frac{\partial\psi}{\partial x}\right)_{i+1/2} = \frac{\psi_{i-1} - 27\psi_i + 27\psi_{i+1} - \psi_{i+2}}{24\Delta x} + O(\Delta x^4), \tag{31}$$

where $x$ can be either $\varphi$ or $\lambda$, $\psi$ is scalar quantity or vector component. However, we use unstaggered grid and need the values of $\frac{\partial\psi}{\partial x}$ to be located at the same points as $\psi$, thus we interpolate $\left(\frac{\partial\psi}{\partial x}\right)_{i+1/2}$ to the integer nodes of the grid using 4-th order Lagrangian interpolation. To calculate divergence and vorticity, 4-th order Lagrangian interpolation is used to obtain vector components at the grid half-nodes. Then second-order locally-conservative formula is applied to calculate the meridional derivative:

$$\frac{1}{a\cos\varphi}\frac{\partial\psi\cos\varphi}{\partial\varphi} = \frac{\psi_{j+1/2}\cos\varphi_{j+1/2} - \psi_{j-1/2}\cos\varphi_{j-1/2}}{a(\sin\varphi_{j+1/2} - \sin\varphi_{j-1/2})} + O(\Delta\varphi^2), \tag{32}$$

the longitudinal derivative is evaluated with Eq. (31) as in the gradient operator.





To account for the variable resolution in latitude, we introduce pseudo-latitude $\varphi'$, such that the grid is equally spaced in $\varphi'$ (following Tolstykh, 2003). Then the meridional derivative $\frac{\partial \psi}{\partial \varphi} = \frac{\partial \psi}{\partial \varphi'} \frac{\partial \varphi'}{\partial \varphi}$. The derivative $\frac{\partial \psi}{\partial \varphi'}$ and inverse mapping factor $\mathcal{M} = \frac{\partial \varphi}{\partial \varphi'}$ are calculated using Eq. (31).

The longitudinal derivatives are calculated in the grid point space, whereas Fourier representation in longitude

$$\psi(\varphi_j, \lambda) = \hat{A}_0(\varphi_j)/2 + \sum_k \left( \hat{A}_k(\varphi_j)\cos(k\lambda) + \hat{B}_k(\varphi_j)\sin(k\lambda) \right) \tag{33}$$

is used to calculate meridional derivatives on the reduced lat-lon grid. Actually, we calculate meridional derivatives of $\hat{A}_k$, $\hat{B}_k$ and then use inverse Fourier transform. If $N_j$ is the number of points at the grid latitude $\varphi_j$, the wave-numbers $k > N_j/2 - 1$ can not be represented, so it is natural to set $\hat{A}_k$, $\hat{B}_k$ and their derivatives to zero at this latitude.

The disadvantage of FFT is the nessecity of global communications in parallel implementation, which could lead to poor
scalability. So we also implement an option to approximate meridional derivtives on the reduced lat-lon grid in grid point space. First we use interpolation along longitude to align $\psi$ and then apply Eq. (31) to obtain half-integer values of derivatives. The similar approach for calculation was applied in (Jablonowski et al., 2009). We have tested this grid point algorithm and there is no impact on the results for the baroclinic test case presented in Sect. 9 as compared with the computations in Fourier space as described above.

Calculation of meridional derivatives near the poles using Eq. (31) requires the values of $\psi$ at virtual latitudes $\varphi_{-1} = -\pi/2 - (\varphi_1 + \pi/2)$ and $\varphi_{\text{Nlat}+1} = \pi/2 + (\pi/2 - \varphi_{\text{Nlat}-1})$. If one continues the meridional line $\lambda$ beyond the pole, it will coincide with $\lambda + \pi$. Thus $\psi(\varphi_{-1}, \lambda) = (-1)^\nu \psi(\varphi_1, \lambda + \pi)$, where $\nu$ is 0 for scalar quantities and 1 for vector components (the latter is due to change of basis vectors orientation after $\pi$ phase shift). It is easy to show that $\hat{A}_k(\varphi_{-1}) = (-1)^{k+\nu} \hat{A}_k(\varphi_1)$, the similar relations take place for $\hat{B}_k(\varphi_{-1})$, $\hat{A}_k(\varphi_{\text{Nlat}+1})$, $\hat{B}_k(\varphi_{\text{Nlat}+1})$.

To evaluate divergence and vorticity at the pole point, one uses the fact that the scalar quantities can have only $\hat{A}_0$ non-zero Fourier coefficient at the pole point to be uniquely defined. The 4-th order accurate Lagrangian interpolation is applied to obtain $\hat{A}_0$ polar values from known $\hat{A}_0$ at adjacent latitudes (including virtual latitudes). Similarly, vector components can have only $\hat{A}_1$, $\hat{B}_1$ non-zero Fourier coefficients at the pole point (this corresponds to the unique value of the vector quantity at the pole point and basis vectors dependent on longitude). Fourier coefficients of gradient components are also interpolated to the pole
point.

The Inherently Mass Conserving (IMC) version of the model (Shashkin and Tolstykh, 2014) also requires calculation of flux-divergence operator $\nabla \cdot (\psi \boldsymbol{V})$, where $\psi$ is some scalar function. For mass conservation $\sum_{i,j} \nabla \cdot (\psi \boldsymbol{V})_{ij} S_{ij} = 0$ over the whole sphere, must hold up to machine precision. This operator is discretized (as described in Tolstykh and Shashkin, 2012) in the flux-form with fourth-order accuracy. In the IMC version, this operator with $\psi = 1$ replaces the divergence operator
described above.

### 5.3 Discretization of the horizontal Laplace operator

The Laplace operator arises in the time-discrete divergence equation (22). In the case of Eq. (22a) derived in a discrete way from momentum equations (3) (which is considered standard in SL-AV20) $\nabla^2$ appears in the implicit part, whereas in the explicit





part it stands as the product of discrete divergence and gradient operators (see Sect. 4 for details). If analytical divergence equation (22b) is used, $\nabla^2$ is present in both implicit and explicit parts of equation.

If the discrete-derived divergence equation is used, the compact finite-difference (CompFD) approximation of the meridional derivatives in the Laplace operator is applied, for their smaller truncation errors. CompFDs discretization require matrix

5  inversion that generaly causes the parallel scalability problems. However, using CompFDs to discretize Laplace operator in the implicit part of divergence equation is not a computational burden, since it is already involved in the matrix inversion problem (see Sect. 4, 6.2).

The Laplace operator is discretized using Fourier representation in longitude. The meridional part of Laplacian is discretized with double application of compact formula for the first derivative:

$$\frac{1}{24}\Big(\frac{\partial\psi}{\partial\varphi}\Big)_{i-1} + \frac{11}{12}\Big(\frac{\partial\psi}{\partial\varphi}\Big)_i + \frac{1}{24}\Big(\frac{\partial\psi}{\partial\varphi}\Big)_{i+1} = \frac{\psi_{i+1/2} - \psi_{i-1/2}}{\Delta\varphi} + O(\Delta\varphi^4), \tag{34}$$

The longitudinal part of $\nabla^2$ is the Fourier image of product of two longitudinal derivatives defined by Eq. (31).

$$\frac{1}{a^2\cos^2\varphi}\frac{\partial^2\exp(ik\lambda)}{\partial\lambda^2} = -\tilde{k}^2\exp(ik\lambda) = -\left(\frac{261\sin(k\Delta\lambda)+\sin(3k\Delta\lambda)-36\sin(2k\Delta\lambda)}{192a\cos\varphi\Delta\lambda}\right)^2\exp(ik\lambda). \tag{35}$$

Further details about compact finite-difference approximation of $\nabla^2$ operator are in Sect. 6.2.

The main objective of using analytically-derived divergence equation (22b) is time-symmetric discretization, which theo-

15 retically leads to better inertia-gravity wave dispersion properties (Caluwaerts et al., 2015). Temporal symmetry means that the same discretization of $\nabla^2$ is used in implicit and explicit parts of the divergence equation. We do not use CompFDs for approximation of Laplace operator to avoid matrix inversion in the explicit part of the equation. The following approximation to $\nabla^2$ is used:

$$\begin{aligned}20\quad(\nabla^2\psi)_{i,j} = &\frac{1}{12a^2\cos^2\varphi_j\Delta\lambda^2}\Big(\psi_{i-2,j}+16\psi_{i-1,j}-30\psi_{i,j}+16\psi_{i+1,j}-\psi_{i+2,j}\Big)+\\ &\frac{1}{a^2\cos\varphi_j\Delta\varphi}\left(\Big(\frac{\partial\psi}{\partial\varphi}\Big)_{i,j+1/2}\cos\varphi_{j+1/2}-\Big(\frac{\partial\psi}{\partial\varphi}\Big)_{i,j-1/2}\cos\varphi_{j-1/2}\right),\quad(36)\end{aligned}$$

$(\partial\psi/\partial\varphi)_{i,j\pm1/2}$ are evaluated using Eq. (31).

### 5.4  Non-linear terms

The calculation of non-linear terms can give rise to non-linear instability, when the interaction of shortest scales leads to the

25 exaggerated contribution to the largest scales. We use spatial averaging to suppress this kind of instability. The following formulae do not have a strict theoretical basis, but they are rather a result of experience in optimizing model performance.

The first type of non-linear term is scalar-by-scalar product, namely $\zeta D$, $D^2$ in vorticity and analytic divergence equations (1),(4):

$$(\zeta D)_{i,j} = \frac{1}{16}\sum_{m=\pm1}\left[\Big(\zeta_{i,j}+\zeta_{i+m,j}\Big)\Big(D_{i,j}+D_{i+m,j}\Big) + \Big(\zeta_{i,j}+\zeta_{i,j+m}\Big)\Big(D_{i,j}+D_{i,j+m}\Big)\right], \tag{37}$$





the term $D^2$ is computed in similar way. Surprisingly, this leads to better medium-range forecast scores as compared to the formula with the increased weight of central term $\zeta_{i,j} D_{i,j}$.

Before computing non-linear terms involving first-order derivatives in $\lambda$ or $\varphi$ (components of $\nabla \ln p_s$, $\partial \dot{\eta}/\partial x$, $\partial T_v/\partial x$), these derivatives are averaged in the direction perpendicular to the direction of differentiation. E.g. $\partial T_v/\partial \lambda$ is averaged in

latitude. The averaging formula is

$$\overline{\left(\frac{\partial \psi}{\partial x}\right)}_j = \frac{1}{2+c}\left(\left(\frac{\partial \psi}{\partial x}\right)_{j-1} + c\left(\frac{\partial \psi}{\partial x}\right)_j + \left(\frac{\partial \psi}{\partial x}\right)_{j+1}\right), \tag{38}$$

where $j$ is index in the direction perpendicular to $x$. The constant $c = 3$ for $T_v$ and $\dot{\eta}$ derivatives, $c = 4$ for $\nabla \ln p_s$ components. The horizontal wind components and temperature are averaged in 2D for computation of $J_\zeta$ Eq. (2), $J_D$ Eq. (5) and terms involving $T_v$ in momentum (3) and thermodynamic (6) equations:

$$\overline{u}_{i,j} = \frac{1}{12}\left(8u_{i,j} + \sum_{m=\pm 1}(u_{i+m,j+m} + u_{i+m,j-m})\right), \qquad \overline{T_v}_{i,j} = \frac{1}{12}\left(8T_{vi,j} + \sum_{m=\pm 1}\left(T_{vi+m,j} + T_{vi,j+m}\right)\right). \tag{39}$$

Equations (37)–(39) are valid in the case of constant latitudinal resolution . To account for the variable resolution, we mention that $\int \psi \mathrm{d}\varphi = \int \psi \mathcal{M} \mathrm{d}\varphi'$ and multiply all $(\zeta_{i,j} + \zeta_{i,j\pm 1})(D_{i,j} + D_{i,j\pm 1})$-type combinations in Eq. (37) and $\psi_{i,j\pm 1}$ terms of Eqs. (38), (39) by $2\mathcal{M}_{j\pm 1/2}/(\mathcal{M}_{j+1/2} + \mathcal{M}_{j-1/2})$.

No averaging is applied for computations of terms involving the coefficients of hybrid coordinate system $A$, $B$, and their

derivatives. Also, no averaging is applied for computation of $R_{\mathrm{moist}}/c_p$ term in thermodynamic equation (6) and hydrostatic equation (10) terms.

## 5.5 Vertical discretization

We use Lorenz-staggered grid (Lorenz, 1960) in the vertical where all variables, except vertical velocity $\dot{\eta}$ are located at integer levels $\eta_l$ (cell centers) and $\dot{\eta}$ is located at half-levels $\eta_{l+1/2}$ (cell interfaces). Integer grid levels are set $\eta_l = (\eta_{l+1/2} + \eta_{l-1/2})/2$,

$\eta_{1/2}$ corresponds to the model top, $\eta_{\mathrm{Nlev}+1/2}$ is at Earth surface. Similarly to $\eta$, the hybrid coordinate system coefficient $A_l = (A_{l+1/2} + A_{l-1/2})/2$ and the same holds for $B_l$, $\Delta \eta_l = \eta_{l+1/2} - \eta_{l-1/2}$ and the same is implied for $\Delta A_l$, $\Delta B_l$ and other variables. The vertical derivatives are approximated with 2-nd order of accuracy $\left(\frac{\partial \psi}{\partial \eta}\right)_l = \frac{\Delta \psi_l}{\Delta \eta_l} + O(\Delta \eta_l^2)$.

It is almost standard for hydrostatic atmospheric models to use Simmons and Burridge (1981) vertical discretization, where the geopotential is calculated at half-levels $\eta_{l+1/2}$ by integrating hydrostatic equation Eq. (10) with mid-point quadrature rule

and then interpolated to the integer levels. It was found that the trapezoidal rule allowing to calculate geopotential directly at the integer levels is more accurate. Eq. (10) is integrated in the following way:

$$\Phi_{\mathrm{Nlev}} = \Phi_s + R_d T_{v\mathrm{Nlev}} \ln \frac{p_s}{p_{\mathrm{Nlev}}}, \tag{40}$$

$$\Phi_{l-1} = \Phi_l + \frac{R_d}{2}\left(T_{vl} + T_{vl-1}\right) \ln \frac{p_l}{p_{l-1}}. \tag{41}$$

The vertical integrals in the time discrete continuity equation (24) and also Eqs. (25), (26) are calculated using mid-point

rule. At the end of each time-step the vertical velocity is recalculated via the diagnostic relation derived from the Eulerian form





of continuity equation (9):

$$\frac{\partial B}{\partial \eta}\frac{\partial p_{\mathrm{s}}}{\partial t} = -\nabla \cdot \left(\frac{\partial p}{\partial \eta}\boldsymbol{V}\right) - \frac{\partial}{\partial \eta}\left(\frac{\partial p}{\partial \eta}\dot{\eta}\right). \tag{42}$$

Integration of Eq. (42) from $\eta_{1/2}$ to $\eta_{L+1/2}$ using mid-point rule and vertical boundary conditions gives

$$\left(B_{L+1/2} - B_{1/2}\right)\frac{\partial p_{\mathrm{s}}}{\partial t} = -\sum_{l=1}^{l=L}\left(A_l p_0 D_l + B_l \nabla \cdot (p_{\mathrm{s}}\boldsymbol{V})_l\right)\Delta\eta_l - \left(\frac{\partial p}{\partial \eta}\dot{\eta}\right)_{L+1/2}. \tag{43}$$

5   The expression for $\frac{\partial p_{\mathrm{s}}}{\partial t}$ can be obtained by setting $L = \mathrm{Nlev}$ in Eq. (43) and using $\dot{\eta}_{\mathrm{Nlev}+1/2}=0$. After substituting this expression back to Eq. (43) for arbitrary $L$, $(\partial p/\partial \eta \cdot \dot{\eta})_{L+1/2}$ can be evaluated. The term $\nabla \cdot (p_{\mathrm{s}}\boldsymbol{V})$ is calculated using standard divergence discretization (see Sect. 5.2), second-order averaging is applied to $p_{\mathrm{s}}$ in latitude (longitude) before computing $p_{\mathrm{s}}u$ ($p_{\mathrm{s}}v$). To retrieve the vertical velocity $\dot{\eta}$ for calculation of upstream trajectories (see Sect. 3) and terms of Eq. (1), $(\partial p/\partial \eta \cdot \dot{\eta})$ is interpolated to integer levels with second-order accuracy and divided by $(p_0\Delta A_l + p_{\mathrm{s}}\Delta B_l)/\Delta\eta_l$ which is proxy for $\partial p/\partial \eta$.

## 10   6   Elliptic problem solution

### 6.1   Wind velocity reconstruction

Following Tolstykh and Shashkin (2012), the definitions of $\zeta = \boldsymbol{k}\cdot\nabla\times\boldsymbol{V}$ and $D = \nabla\cdot\boldsymbol{V}$ are applied to reconstruct the horizontal wind from known vertical component of relative vorticity and horizontal divergence. Fourier representation in longitude Eq. (33) is used. The relations for $u$ and $v$ 0-th Fourier component are:

$$15 \quad \frac{\partial \hat{A}_0^u \cos\varphi}{\partial \varphi} = \hat{A}_0^\zeta a \cos\varphi, \tag{44}$$

$$\frac{\partial \hat{A}_0^v \cos\varphi}{\partial \varphi} = \hat{A}_0^D a \cos\varphi. \tag{45}$$

These relations are integrated in latitude with the compact formula (Lele, 1992):

$$\frac{1}{24}\left(\frac{\partial\psi}{\partial\varphi}\right)_{j-1} + \frac{11}{12}\left(\frac{\partial\psi}{\partial\varphi}\right)_{j} + \frac{1}{24}\left(\frac{\partial\psi}{\partial\varphi}\right)_{j+1} = \frac{\psi_{j+1/2} - \psi_{j-1/2}}{\Delta\varphi} + O(\Delta\varphi^4), \tag{46}$$

where $\hat{A}_0^\zeta \cos\varphi$ ($\hat{A}_0^D \cos\varphi$) is substituted for $\partial\psi/\partial\varphi$ in the left hand side and $\hat{A}_o^u \cos\varphi$ ($\hat{A}_o^v \cos\varphi$) is substituted for $\psi$ in the

20   right hand side. The resulting Fourier coefficients for horizontal wind components at half grid latitudes are interpolated to the integer grid latitudes using 6-th order compact interpolation (Lele, 1992).

  The $k$-th Fourier coefficients of horizontal wind components are determined solving

$$\begin{cases} -k\hat{A}_k^v - \frac{\partial \hat{B}_k^u \cos\varphi}{\partial\varphi} = \hat{B}_k^\zeta a \cos\varphi, \\ k\hat{B}_k^u + \frac{\partial \hat{A}_k^v \cos\varphi}{\partial\varphi} = \hat{A}_k^D a \cos\varphi, \end{cases} \tag{47}$$

$$\begin{cases} -k\hat{A}_k^u + \frac{\partial \hat{B}_k^v \cos\varphi}{\partial\varphi} = \hat{B}_k^D a \cos\varphi, \\ k\hat{B}_k^v - \frac{\partial \hat{A}_k^u \cos\varphi}{\partial\varphi} = \hat{A}_k^\zeta a \cos\varphi, \end{cases} \tag{48}$$



where $\partial\psi/\partial\varphi$ is approximated using Numerov's scheme:

$$\frac{1}{6}\Big(\frac{\partial\psi}{\partial\varphi}\Big)_{j-1}+\frac{2}{3}\Big(\frac{\partial\psi}{\partial\varphi}\Big)_{j}+\frac{1}{6}\Big(\frac{\partial\psi}{\partial\varphi}\Big)_{j+1}=\frac{\psi_{j+1}-\psi_{j-1}}{2\Delta\varphi}+O(\Delta\varphi^4). \tag{49}$$

If $\hat{A}_k^\psi$, $\hat{B}_k^\psi$ are $\mathrm{Nlat}+1$-element columns, with $j$-th element representing $\hat{A}_k^\psi$, $\hat{B}_k^\psi$ at $j$-th grid latitude, then system (47) can be written as:

$$\begin{cases} -k\hat{\boldsymbol{A}}_k^v-\frac{1}{2\Delta\varphi}\mathbf{M}^{-1}\delta\mathbf{C}\hat{\boldsymbol{B}}_k^u=a\mathbf{C}\hat{\boldsymbol{B}}_k^\zeta, \\ k\hat{\boldsymbol{B}}_k^u+\frac{1}{2\Delta\varphi}\mathbf{M}^{-1}\delta\mathbf{C}\hat{\boldsymbol{A}}_k^v=a\mathbf{C}\hat{\boldsymbol{A}}_k^D, \end{cases} \tag{50}$$

where $\mathbf{C}$ is the diagonal matrix with $\mathbf{C}_{jj}=\cos\varphi_j$, $j$-th component of column $\delta\hat{\boldsymbol{A}}$ is $\hat{A}_{j+1}-\hat{A}_{j-1}$, $\mathbf{M}$ is matrix with diagonals $(1/6,2/3,1/6)$. We multiply the system of Eqs. (50) by matrix $\mathbf{M}$ from the left and rewrite it for $(\hat{A}_k^v,\hat{B}_k^u)_j^T$ pairs, $j\in[0,\mathrm{Nlat}]$ that results in $2\times2$ block tri-diagonal system of equations. The same operations repeated for system (48) yields $2\times2$ block tri-diagonal system of equations for $(\hat{A}_k^u,\hat{B}_k^v)_j^T$. The details of this solver are given in Tolstykh and Shashkin (2012). As shown in Caluwaerts et al. (2015), this solver alleviates the problems associated with non-temporal symmetric discretization. Also, it avoids the solution of Poisson problem on the sphere which requires certain care due to the non-trivial kernel of Laplace operator.

## 6.2 Helmholtz problem solution

The discrete Helmholtz problem for $\psi$ – the $k$-th Fourier harmonic of arbitrary quantity can be written in the column-matrix notation of Sect. 6.1 as $\mathbf{L}\psi+\mu^2\psi=\boldsymbol{R}$, where $\mathbf{L}$ is a discrete Laplace operator, $\mu^2$ is some scalar, $\boldsymbol{R}$ is $k$-th Fourier harmonic of RHS. When divergence equation (22a) derived from time-discrete momentum equation (3) is used, the meridional part of $\nabla^2$ operator is approximated by double application of compact finite-difference formula for the first derivative Eq. (34) and Eq. (35) is used for the approximation of longitudinal part of Laplacian. The resulting 4-th order accurate Helmholtz problem discretization is

$$\frac{1}{a^2}\mathbf{C}^{-1}\mathbf{M}^{-1}\delta_{1/2}\Big(\mathbf{C}_{1/2}\mathbf{M}_{1/2}^{-1}\delta\psi\Big)+\Big(\mu^2-\frac{1}{a^2}\mathbf{C}^{-2}\tilde{k}^2\Big)\psi=\boldsymbol{R}, \tag{51}$$

where operator $\delta_{1/2}$ acts on quantities defined at half-integer grid points, $j$-th component of $\delta_{1/2}\psi$ is $\psi_{j+1/2}-\psi_{j-1/2}$, $\delta$ acts on quantities defined at integer grid points, $j$-th component of $\delta\psi$ is $\psi_{j+1}-\psi_j$, $\mathbf{M}$ and $\mathbf{M}_{1/2}$ are tri-diagonal matrices with diagonals $(1/24,11/12,1/24)$ acting on quantities defined at integer and half-integer points respectively. Matrix $\mathbf{C}$ is the same as in Eq. (50), matrix $\mathbf{C}_{1/2}$ is diagonal with $j$-th diagonal element $\cos\varphi_{j+1/2}$.

Following Tolstykh (2002), Eq. (51) is multiplied by $\mathbf{MC}$ from the left side and reformulated using auxiliary variable $\boldsymbol{z}=\mathbf{M}_{1/2}^{-1}\delta\psi$:

$$\frac{1}{a^2}\delta_{1/2}\Big(\mathbf{C}_{1/2}\boldsymbol{z}\Big)+\mathbf{M}\Big(\mathbf{C}\mu^2-\frac{1}{a^2}\mathbf{C}^{-1}\tilde{k}^2\Big)\psi=\mathbf{MC}\boldsymbol{R}, \tag{52}$$

$$\mathbf{M}_{1/2}\boldsymbol{z}=\delta\psi. \tag{53}$$





Then (52) is rewritten for the pairs $(\psi_j, z_{j+1/2})^T$, $j = [0, \mathrm{Nlat}]$ that results in $2 \times 2$ block tri-diagonal system as in Sect. 6.1.

Approximation (36) to $\nabla^2$ operator is used when the analytically derived divergence equation (22b) is employed in the system of equations (27), (28). This approximation leads to 5-diagonal system of equations for $\psi$ components which is solved by 5-diagonal Gauss elimination.

## 7  Dissipation mechanisms

### 7.1  Fourth order hyper-diffusion

Non-linear interactions in the complex large-scale atmospheric flows results in the generation of progressively smaller eddies until these eddies are converted to heat by molecular viscosity at the scales of order of $1$cm. Such scales are far beyond the affordable resolution of atmospheric models. Therefore, the parameterization of unresolved scales interactions and dissipation
is needed to avoid accumulation of energy at smaller scales. Such parameterizations are often considered as the integral part of the atmospheric model dynamical core (Williamson, 2007). The SL-AV20 model includes the implicit in time fourth-order diffusion implemented in Fourier space with the finite volume representation in latitude (Tolstykh, 1997). This algorithm, allowing variable resolution in latitude, is a generalization of that one presented in (Li and Bates, 1994). We start with the equation

$$\psi_f^{n+1} = \psi^{n+1} - K\Delta t \nabla^4 \psi_f^{n+1} \tag{54}$$

where $\psi^{n+1}$ is one of $D^{n+1}$, $T_v^{n+1}$, $\zeta^{n+1}$, $\dot{\eta}^{n+1}$ and $\psi_f^{n+1}$ is the filtered quantity. Equation (54) uses implicit time-stepping that allows us to circumvent severe limitation on $K$ near the grid poles, $\Delta t$ is the same as in the rest of the model dynamics.

SL-AV20 uses variable resolution grids, so the filter should not excessively dissipate smaller scale features in high-resolution regions. However, these features should be damped out before they spread to the low resolution regions where they cannot be
resolved. We use anisotropic diffusion coefficient that varies with latitude, the $\nabla^4$ is substituted for $\nabla \cdot (\mathbf{K}\nabla^3\psi)$ to preserve local conservation, $\mathbf{K} = \mathrm{diag}(K_\lambda, K_\varphi)$. To facilitate numerical solution, Eq. (54) is reformulated as

$$
\begin{aligned}
\psi_f^{n+1} &= \psi^{n+1} - \Delta t \nabla \cdot \mathbf{K}\nabla\xi, \\
\xi &= \nabla^2 \psi_f^{n+1}.
\end{aligned}
\tag{55}
$$

We use second order finite-volume formula to approximate the latitudinal part of $\nabla \cdot \mathbf{K}\nabla$ and $\nabla^2$ operators:

$$
\left(\nabla \cdot \mathbf{K}\nabla\xi\right)_\varphi = \frac{1}{a^2 \Delta(\sin\varphi)_j}\left(K_{\varphi_{j+1/2}} \cos\varphi_{j+1/2} \frac{\xi_{j+1} - \xi_j}{\Delta\varphi_{j+1/2}} - K_{\varphi_{j-1/2}} \cos\varphi_{j-1/2} \frac{\xi_j - \xi_{j-1}}{\Delta\varphi_{j-1/2}}\right) + O(\Delta\varphi^2)
\tag{56}
$$

where $\varphi_{j+1/2} = 1/2(\varphi_j + \varphi_{j+1})$, $\Delta\varphi_{j+1/2} = \varphi_{j+1} - \varphi_j$, $\Delta(\sin\varphi)_j = \sin\varphi_{j+1/2} - \sin\varphi_{j-1/2}$. Longitudinal part of $\nabla \cdot \mathbf{K}\nabla$ and $\nabla^2$ is approximated in Fourier space as the Fourier image of second-order discretization.



The system Eqs. (55) using (56) can be written for zonal wave-number $k$ as

$$\mathbf{A} \begin{pmatrix} \xi \\ \psi_f^{n+1} \end{pmatrix}_{j-1} + \mathbf{B} \begin{pmatrix} \xi \\ \psi_f^{n+1} \end{pmatrix}_{j} + \mathbf{C} \begin{pmatrix} \xi \\ \psi_f^{n+1} \end{pmatrix}_{j+1} = \frac{a^2 \Delta(\sin\varphi)_j}{\Delta t} \begin{pmatrix} \psi^{n+1} \\ 0 \end{pmatrix}_{j}, \tag{57}$$

$$\mathbf{B} = \begin{pmatrix} -K_{\varphi\, j-1/2}\frac{\cos\varphi_{j-1/2}}{\Delta\varphi_{j-1/2}} - K_{\varphi\, j+1/2}\frac{\cos\varphi_{j+1/2}}{\Delta\varphi_{j+1/2}} - \frac{\tilde{k}^2 K_{\lambda\, j}\Delta(\sin\varphi)_j}{\cos^2\varphi_j} & \frac{a^2 \Delta(\sin\varphi)_j}{\Delta t} \\ -a^2\Delta(\sin\varphi)_j & -\frac{\cos\varphi_{j-1/2}}{\Delta\varphi_{j-1/2}} - \frac{\cos\varphi_{j+1/2}}{\Delta\varphi_{j+1/2}} - \frac{\tilde{k}^2\Delta(\sin\varphi)_j}{\cos^2\varphi_j} \end{pmatrix}, \tag{58}$$

$$\mathbf{A} = \begin{pmatrix} K_{\varphi\, j-1/2}\frac{\cos\varphi_{j-1/2}}{\Delta\varphi_{j-1/2}} & 0 \\ 0 & \frac{\cos\varphi_{j-1/2}}{\Delta\varphi_{j-1/2}} \end{pmatrix}, \mathbf{C} = \begin{pmatrix} K_{\varphi\, j+1/2}\frac{\cos\varphi_{j+1/2}}{\Delta\varphi_{j+1/2}} & 0 \\ 0 & \frac{\cos\varphi_{j+1/2}}{\Delta\varphi_{j+1/2}} \end{pmatrix} \tag{59}$$

where $-\tilde{k}^2 = -(1-\cos k\Delta\lambda)/\Delta\lambda^2$ is the Fourier image of longitudinal part of discrete Laplace operator. This system is solved using $2\times 2$ block tri-diagonal version of Gauss elimination.

## 7.2 Sponge layer

The divergence damping at the vertical levels close to model top is used to avoid spurious reflection of vertically propagating waves from the upper rigid lid ($\dot\eta = 0$ at $\eta_{1/2}$ boundary condition). The term $-\vartheta(\eta)D^{n+1}$ is included into the RHS of discrete divergence equation (22). Implicit time-stepping allows us to feel free in choosing damping coefficient $\vartheta$ and does not complicate the solution of system (27), (28).

## 8 Parallel implementation

SL-AV20 model uses hybrid distributed-shared memory parallelism based on combination of MPI and OpenMP technologies. Each MPI process performs computations in the band of grid latitudes during the first phase of the time-step that includes upstream trajectory computations, interpolation and combination of known terms into RHS of SI system of equation (see Sect. 4). OpenMP threads are used to parallelize longitude loops thus dividing the latitude belt into a number of parts.

The second phase of SL-AV20 time-step consists of the solution of SI system of equations, application of 4-th order hyper-diffusion (Sect. 7) and reconstruction of wind velocity from vorticity and divergence (Sect. 6.1). This phase is mostly the solution of elliptic problems. To apply the direct solvers described in Sects. 6, 7, it is convenient to gather $k$-th Fourier coefficients from all grid latitudes in the memory of specific MPI-process. Therefore, the second phase of the SL-AV20 time-step is preceded by data transposition. Each MPI-process performs computations within the set of longitude wave-numbers from pole to pole. OpenMP parallelization of loops in vertical is applied.

To balance work-load of MPI-processes, the widths of corresponding latitude bands are calculated accounting for the grid reduction, so each process treats approximately the same number of grid points. There is an additional constraint of current implementation that all grid points for a given latitude should be located at the same processor. There is no load balance problem in Fourier-space computations, as the processor partition is the same as for regular grid. One can additionally adjust load balance by taking into account that some parameterizations only work in some latitude band.





The data transpositions before and after the solution of elliptical problems require global communications between the processors and is a future scalability bottleneck, so we work on scalable iterative grid-point solvers for elliptic problems. It is known that iterative solvers for problems described in Sects. 4, 6, 7 can scale up to tens of thousands processors (Müller and
Scheichl, 2014). The semi-Lagrangian advection code is also known to scale at $10^4$ processors (White III and Dongarra, 2011).

Currently, full SL-AV20 code with parameterizations runs at 3024 cores with 70% efficiency, at 4536 cores with 63% efficiency, and at 9072 cores with 45 % efficiency when using the grid of 3024 by 1513 points in longitude and latitude respectively (see Fig. 1). This grid corresponds to 13 km resolution at the equator and has 51 levels in vertical. Only 288 cores are needed to meet HMCR operational NWP requirements (20 minutes per 24 model hours) at the current operational
horizontal grid of 1600 by 865 points.

## 9   Numerical experiments

Verification of atmospheric models dynamical cores with idealized adiabatic (moist-adiabatic, quasi-adiabatic) testcases such as Held and Suarez (1994), Jablonowski and Williamson (2006b), Reed and Jablonowski (2012), Thatcher and Jablonowski (2016) gained popularity last years. The reason is that various aspects of model numerics can be evaluated in easier way
using idealized setup than real atmospheric flow with all its complexity. Also, the numerical solutions to idealized tests are independent of sub-grid scale processes parameterizations.

The full verification of SL-AV20 dynamical core with all its options can hardly be presented in one article. Here we concentrate on the impact of the horizontal grid aspects on the numerical solution, namely, the grid reduction and resolution refinement (see Sect. 5.1). The results presented below are obtained with baseline NWP setup of SL-AV20 dynamical core
with no inherent mass-conservation and divergence equation (22a) derived from the discrete form of momentum equation (3).

The grids with $400 \times 250$, $800 \times 500$, $1600 \times 865$ points in longitude and latitude respectively and 28 levels in vertical are used in this study. The first and last horizontal grids correspond to the old and new SL-AV operational NWP configurations at HMCR. The algorithm proposed by Fadeev (2013) is used to obtain variable resolution and reduced variants of these grids. The latitude grid spacing is refined in the Northern hemisphere mid-latitudes. The reduced grids used in this study have about
25% less points than the source regular grids. As we learned from shallow water model (Tolstykh and Shashkin, 2012) and advection (Shashkin et al., 2016) experiments, 25% reduction is almost the greatest reduction that does not significantly damage the solution. The characteristics of grids used in this study are summarized in Fig. 2.

The time-step for all variants of $400 \times 250$ grid is $1200\,\mathrm{s}$ and reduced by the factor $c = \frac{400}{\mathrm{Nlon}}$ for finer grids (Nlon is the number of grid points along the equator). The basic $\nabla^4$ hyper-diffusion coefficients in $400 \times 250$ grid experiments are $1.5 \times 10^{15}\mathrm{m^4 s^{-1}}$
for relative vorticity and temperature and $1.9 \times 10^{15}\mathrm{m^4 s^{-1}}$ for divergence, the hyper diffusion coefficients for finer grids are reduced by the factor $c^3$. The latitudinal component of hyper-diffusion coefficient (see Sect. 7) varies with latitudinal grid spacing $K_\varphi = (\Delta\varphi/\Delta\varphi')^3 K_0$, where $\Delta\varphi' = \frac{2\pi}{\mathrm{Nlat}}$ and $K_0$ is the basic coefficient value for the given grid.



## 9.1 Baroclinic instability test case

In the baroclinic instability test case by Jablonowski and Williamson (2006b), the small zonal wind perturbation of the geostrophically balanced background state is placed in the Northern hemisphere mid-latitudes. After initial adjustment, the perturbation causes exponential growth of the baroclinic wave (days 1-7) until it breaks down (days 7-10) and transforms to the fully chaotic solution in the course of other 20 days. The evolution of test case solution (obtained by different models) is illustrated in details in Jablonowski and Williamson (2006a).

As can be seen from figures in Jablonowski and Williamson (2006a) and Wan et al. (2013), the surface pressure and temperature fields rapidly converge to the smooth large scale pattern with the resolution increase. Contrary, the relative vorticity field of breaking wave consists of filaments that become more and more twisted and thinner with greater amplitude when the grid resolution is increased. Each step of grid refinement reveals new smaller scale features in this field.

Our particular interest is the net effect of increasing latitudinal and reducing longitudinal resolution in the zone of interest i.e. Northern hemisphere mid-latitudes. Figure 3 presents the comparison of day 9 850 $\mathrm{hPa}$ relative vorticity field between numerical solutions on regular lat-lon grids with constant grid spacing in latitude and reduced lat-lon grids with variable latitudinal resolution. The absolute values of vorticity and its gradients are greater and the inner structure of the vortices is more developed in the solutions at $400 \times 250$ and $800 \times 500$ variable resolution reduced grids as compared to the solutions obtained at corresponding regular grids. The improvement in $1600 \times 865$ grid solution is less obvious that can be partly attributed to the convergence. Slightly stronger development of the leftmost and middle vortices can be noticed in the variable resolution solution after closer consideration. However, some subtle details of the rightmost vortex are lost due to the grid reduction (see discussion below).

To isolate the effect of the grid reduction, we compare the day 9 850 $\mathrm{hPa}$ relative vorticity snapshots from the variable resolution grid solutions with and without reduction of longitudinal resolution (Fig. 4). Except the leading vortex on $1600 \times 865$ grid, the difference between the solutions using the reduced and non-reduced grids can hardly be found. Therefore, the reduction of longitudinal resolution does not lead to the degradation of solution. At the same time, one can hope that reproduction of mid-latitude pressure systems dynamics can be improved using variable resolution in latitude. Nevertheless, it should be noted that the effect of latitudinal resolution refinement is weaker than the effect of increasing resolution of the basic grid.

One can see in Figs. 3, 4 that SL-AV20 solutions are smooth and no evidence of grid-scale noise or any other type of noise can be found. This does not compromise the reproduction of smaller scale details, gradient strength and amplitude of the relative vorticity field which agree well with the solutions of comparable resolution (ICON (Zangl et al., 2015), ECHAM (presented in Wan et al., 2013), CAM-FV, CAM-EUL, CAM-SLD, GME (compared in Jablonowski and Williamson, 2006a). The surface pressure error norms with respect to CAM-SLD reference solution (not shown) are below the uncertainty level (Jablonowski and Williamson, 2006b), similar to the previous version of SL-AV model (Shashkin and Tolstykh, 2014) .





## 9.2 Held-Suarez test

The goal of Held and Suarez (1994) experiment is to verify statistical properties of the atmospheric model circulation driven by highly idealized forcing – the relaxation of temperature to the equilibrium profile and Rayleigh friction near the surface. The resulting atmospheric regime is a balance between non-uniform heating, polewards transport of heat and momentum by baroclinic eddies and destruction of the momentum in the boundary layer. Thus, the accurate representation of baroclinic motions in the mid-latitudes plays a crucial role for the test results.

Both lat-lon grid reduction and local increase of latitudinal resolution can modify the intensity of eddy heat and momentum fluxes and thus alter the net circulation characteristics. Figure 5, panels (a) – (f) presents the 1000 days time-zonal averaged zonal wind speed, temperature, eddy heat and momentum fluxes, eddy kinetic energy and temperature variance. Shading shows the solution at $400 \times 250$ reduced grid with variable resolution in latitude, regular grid solution with constant latitudinal resolution is shown by contours. One can note a slight increase of Northern hemisphere eddy momentum flux (near 300 hPa) and temperature variance (near 900 hPa) maximum values. The eddy heat flux in the Southern hemisphere at 300 hPa is slightly degraded in the variable resolution solution that can be co-effect of grid reduction and increased latitudinal grid spacing. The color-levels of eddy kinetic energy to the North of $45°$N in Fig 5, panel (e) are extended polewards as compared to the corresponding contours of regular grid solution. That means more intensive eddy motions at high latitudes in the variable resolution reduced grid solution.

The standalone effect of grid reduction can be evaluated from panels (g) – (l) in Fig. 5. The shading shows the difference between constant latitudinal resolution regular and reduced grid solutions (regular minus reduced), the contours of regular grid solution are overlaid for reference. The majority of features depicted in these panels are explained by the shift of jet-stream cores which is rather the result of internal low-frequency variability of test solution (Wan et al., 2008) than a consequence of grid reduction. We can find no significant decrease in polewards eddy fluxes due to suppression of the shortest-scale motions in the regions with reduced longitudinal resolution.

Figure 6 presents 500 hPa kinetic energy spectra from three numerical solutions (regular, reduced, reduced with variable resolution in latitude) in Northern hemisphere mid-latitudes (left panel) and high-latitudes (right-panel). To obtain the corresponding spectra, $u$ and $v$ are multiplied by window-function equal to 1 in $30°$N–$60°$N (to the North of $70°$N in the case of high-latitudes spectrum) and $10°$-width smooth transition to the zero value outside. The lines corresponding to the regular and reduced grid solutions are almost identical in mid-latitudes for all wave-numbers, whereas variable resolution solution shows more intensive smaller scale motions. The effect of variable resolution in high latitudes also outweighs the effect of grid reduction.

## 10 Conclusions

We have described the dynamical core of SL-AV20 (semi-Lagrangian Absolute Vorticity) atmospheric model thus summarizing all recent developments in the model numerics (Fadeev, 2013; Tolstykh and Shashkin, 2012; Shashkin and Tolstykh, 2014; Shashkin et al., 2016). The model is applied to operational global numerical weather prediction with 20km resolution over





Russia. Also, its lower resolution configurations will be soon applied for probabilistic long-range forecasts and is the base for development of the new atmospheric component of INMCM Earth system model (Volodin et al., 2010).

The main features of the model dynamical core are the vorticity-divergence formulation at the unstaggered grid, reduced

latitude-longitude grid with variable resolution in latitude, using high-order finite-difference formulae, semi-Lagrangian semi-implicit discretization. The number of grid points in longitude at near-polar latitudes for the reduced grids constructed with the algorithm by Fadeev (2013) is an order of magnitude smaller than the number of points at the equator.

The effects of the grid reduction and variable latitudinal resolution on numerical solutions are investigated with two idealized tests – the Jablonowski and Williamson (2006b) determenidtic case and Held and Suarez (1994) long-run experiment. The

results agree with other published model solutions. It is shown that the reduction in number of grid points by 25% leads to only moderate damping of the shortest longitudinal waves and does not degrade the solution. At the same time, variable resolution in latitude allows to refine grid in the region of interest and thus improve the solution accuracy. Particularly, increasing latitudinal resolution in the Northern hemisphere results in sharper gradients and stronger development in the relative vorticity field during unstable baroclinic wave breaking in Jablonowski and Williamson (2006b) test and more intensive mid-latitude eddy motions

in the Held and Suarez (1994) test. In all tests, the evident positive effect of increase in latitudinal resolution outweighs the slight negative effect of grid reduction.

Currently, the experimental SL-AV20 configuration with approximately 13km horizontal resolution can use up to 9000 cores. This is sufficient in a short term perspective but will become a bottleneck after some years once exascale computer architectures will be implemented. So far, we use Fast Fourier transforms in some parts of the dynamical core. Hence data transpositions

(global communications between processors) are needed in parallel implementation. We work on scalable iterative grid-point solvers for elliptic problems which are known to scale up to tens of thousands processors (Müller and Scheichl, 2014). Also, we have successfully tested computation of meridional derivatives at the reduced grid in the grid-point space. These developments will help to avoid the use of FFT in the model code thus moving scalability limit further.

## 11   Code availability

The code of SL-AV20 dynamical core is available for research purposes on request, please contact Mikhail Tolstykh tolstykh@m.inm.ras.ru .

## Appendix A:  List of notations

$a$  – mean Earth's radius

$A(\eta)$, $B(\eta)$  – coefficients of Simmons and Burridge (1981) hybrid vertical coordinate

$\hat{A}_k^\psi$, $\hat{B}_k^\psi$  – $k$-th zonal wave number coefficients of $\psi$ Fourier representation in longitude

$c_{pd}$, $c_p$  – heat capacity of dry and moist air under constant pressure





$D$ – horizontal divergence

$f$ – the Coriolis parameter

$F_\psi$ – source/sink of $\psi$ due to subgrid/diabatic processes

$\quad G$ – linear geopotential term

$i, j$ – longitude and latitude grid-point indices

$J_\zeta, J_D$ – Jacobian-like terms of absolute vorticity and divergence equations (1, 4)

$\boldsymbol{k} - \boldsymbol{r}/a$ – vertical unit vector

$k, \tilde{k}$ – zonal wave-number, the modified zonal wave-number

$\quad l$ – vertical level index

$\mathcal{M}$ – inverse mapping factor $\partial\varphi/\partial\varphi'$

Nlat, Nlon, Nlev – number of grid points in latitude, longitude and vertical respectively

$p, p_0, p_s$ – pressure, constant reference pressure, surface pressure respectively

$\bar{p}, \bar{p}_s$ – reference pressure profile, reference surface pressure

$\quad p_s^{ref}$ – orography dependent reference surface pressure

$q, q_i$ – specific concentraions of water vapour and liquid (solid) water species

$\boldsymbol{r}$ – vector pointing from the center of the sphere to the given point on its surface

$R_d, R_v, R_{\text{moist}}$ – gas constants of dry air, water vapour and moist air respectively

$R_\zeta, \boldsymbol{R_V}, R_D, R_T, R_P$ – RHS of time-discrete absolute vorticity (19), momentum, divergence (22b), thermodynamic (23) and

$\qquad$ continuity (24) equations respectively

$\mathfrak{R}$ – the rotation matrix

$\dot{s}$ – analogue of vertical velocity used in thermodynamic and continuity equations (9,6)

$S_{ij}$ – horizontal square of $i,j$-th vertical column of computational cells

$T, T_v$ – temperature, virtual temperature

$\bar{T}$ – constant reference temperature



$\boldsymbol{V} = (u, v)$  – horizontal wind

$\mathcal{V}_{ijl}$  – computational cell

$\mathcal{V}(t)$  – Lagrangian volume

$\gamma(\eta)\Phi_{\mathrm{s}}$  – Ritchie and Tanguay (1996) correction term for thermodynamic equation (6)

$\Delta\lambda_j$  – grid spacing in longitude at $j$-th grid latitude

$\zeta$  – vertical component of relative vorticity

$\eta$  – hybrid vertical coordinate of Simmons and Burridge (1981)

$\dot{\eta}$  – vertical speed in $\eta$ coordinate

$\lambda$  – longitude

$\varphi$, $\varphi_j$  – latitude, j-th grid latitude

$\varphi'$  – pseudo-latitude

$\Phi$  – geopotential

$\Phi_s$  – surface geopotential

$\psi$  – arbitrary quantity

$\boldsymbol{\Omega}$  – Earth's angular velocity

$(\psi)_*$  – $\psi$ is evaluated at the departure point of upstream trajectory

$(\psi)_{\mathcal{V}_{ijl}}$, $(\psi)_{S_{ij}}$  – average of $\psi$ over $\mathcal{V}_{ijl}$-cell or $S_{ij}$-horizontal square

**Appendix B:  Right-hand side of SI system**

The right-hand sides of the discrete in time equation formulated in Sect. 4.1 are as follows:

$$R_\zeta = \zeta_*^n + f_* - \frac{\Delta t}{2}\left[\left(fD\right)_*^n + \left(\zeta D\right)_*^{(n+1)e} + \left(\zeta D\right)^n + J_{\zeta_*}^{(n+1)e} + J_\zeta^{\,n} + F_\zeta^n\right], \tag{B1}$$

where departure-point Coriolis term $f_* = 2|\Omega|\sin\varphi_*$, $\varphi_*$ is the latitude of departure point, $\left(fD\right)_*^n = f_* D_*^n$, $D_*^n$ is divergence interpolated to the departure point.



$$
\boldsymbol{R_V} = -2\boldsymbol{\Omega} \times \boldsymbol{r}^{n+1} + \frac{\Delta t}{2}\Big(\epsilon\nabla G^n - \nabla G_N^n\Big)
$$
$$
+ \Re\left[\boldsymbol{V}^n + 2\boldsymbol{\Omega} \times \boldsymbol{r}^n + \frac{\Delta t}{2}\Big(-(1+\epsilon)\nabla G^n + \epsilon\nabla G^{(n+1)e} - \nabla G_N^{(n+1)e}\Big) + \boldsymbol{F_V}^n\right]_*, \quad \text{(B2)}
$$

$$
R_D = \frac{\Delta t}{2}\Big(\epsilon\nabla^2 G^n + N_D^n\Big) + D_*^n + \frac{\Delta t}{2}\Big((f\zeta)^n - (1+\epsilon)\nabla^2 G^n + \epsilon\nabla^2 G^{(n+1)e} + N_D^{(n+1)e} + F_D^n\Big)_*, \quad \text{(B3)}
$$

$$
N_D = -D^{n\,2} - R_d\nabla\cdot\Big((T_v - \bar{T})\nabla\ln p_\mathrm{s}\Big) - \frac{u}{a}\frac{\partial f}{\partial\varphi} + J_D. \quad \text{(B4)}
$$

The right-hand side of discretized in time thermodynamic equation (23):

$$
R_T = -\frac{R_d\bar{T}}{c_{pd}}\frac{\epsilon}{2}\Delta t\frac{\bar{p}_\mathrm{s}}{Ap_0 + B\bar{p}_\mathrm{s}}\dot{s}^n + \frac{\Delta t}{2}N_T^n +
$$

$$
(T_v^n)_* - \frac{R_d\bar{T}}{c_{pd}}\left[\frac{B\bar{p}_\mathrm{s}}{Ap_0 + B\bar{p}_\mathrm{s}}\ln p_{\mathrm{s}\,*}^n - \frac{\Delta t}{2}\left(\frac{\bar{p}_\mathrm{s}}{Ap_0 + B\bar{p}_\mathrm{s}}\Big((1+\epsilon)\dot{s}^n - \epsilon\dot{s}^{(n+1)e}\Big)\right)\right]_* + \frac{\Delta t}{2}N_{T\,*}^{(n+1)e} + F_{T_v\,*}^n \quad \text{(B5)}
$$

$$
N_T = \left(\frac{R_{\mathrm{moist}}T_v}{c_p}\frac{p_\mathrm{s}}{Ap_0 + Bp_\mathrm{s}} - \frac{R_d\bar{T}}{c_{pd}}\frac{\bar{p}_\mathrm{s}}{Ap_0 + B\bar{p}_\mathrm{s}}\right)\left(\dot{s} - B\frac{(\partial A/\partial\eta)p_0 + (\partial B/\partial\eta)p_\mathrm{s}}{p_\mathrm{s}(\partial B/\partial\eta)}D - B\Big(\frac{\partial B}{\partial\eta}\Big)^{-1}\dot{s}\right)
$$
$$
+ \gamma(\eta)\boldsymbol{V}\cdot\nabla\Phi_\mathrm{s} + \dot{\eta}\frac{\partial\gamma(\eta)}{\partial\eta}\Phi_\mathrm{s}, \quad \text{(B6)}
$$

15 the terms of $N_T$ proportional to $B(\partial B/\partial\eta)^{-1}$ which arise from substitution of $\mathrm{d_H}/\mathrm{d}t\ln p_\mathrm{s}$ from continuity equation (9) are set 0 when $(\partial B/\partial\eta) = 0$. Actually, $(\partial B/\partial\eta) = 0$ only when $B = 0$ and thermodynamic equation (6) transforms to pure isobaric coordinate form that does not contain $\mathrm{d_H}/\mathrm{d}t\ln p_\mathrm{s}$ at all.

$$
R_P = \frac{\Delta t}{2}\left(\epsilon\frac{(\partial A/\partial\eta)p_0 + (\partial B/\partial\eta)\bar{p}_\mathrm{s}}{\bar{p}_\mathrm{s}}D^n + \epsilon\frac{\partial\dot{s}^n}{\partial\eta} + N_P^n\right) + \frac{\partial B}{\partial\eta}\ln p_{\mathrm{s}\,*}^n
$$

$$
+ \frac{\Delta t}{2}\left[\frac{(\partial A/\partial\eta)p_0 + (\partial B/\partial\eta)\bar{p}_\mathrm{s}}{\bar{p}_\mathrm{s}}\Big(-(1+\epsilon)D^n + \epsilon D^{(n+1)e}\Big) - (1+\epsilon)\frac{\partial\dot{s}^n}{\partial\eta} + \epsilon\frac{\partial\dot{s}^{(n+1)e}}{\partial\eta} + N_P^{(n+1)e}\right]_*, \quad \text{(B7)}
$$

$$
N_P = \left(\frac{(\partial A/\partial\eta)p_0 + (\partial B/\partial\eta)\bar{p}_\mathrm{s}}{\bar{p}_\mathrm{s}} - \frac{(\partial A/\partial\eta)p_0 + (\partial B/\partial\eta)p_\mathrm{s}}{p_\mathrm{s}}\right)D, \quad \text{(B8)}
$$





## Appendix C: SI system matrices

Following column-matrix notation introduced at the end of Sect. 4.1, we define $\boldsymbol{T}$, $\boldsymbol{S}$, $\boldsymbol{P}$ as vectors with $k$-th component representing $T_{v\,k}^{n+1}$, $\dot{s}_k^{n+1}$ and $\ln p_{\rm s}^{n+1}$ 2D fields respectively, and write Eqs. (23), (25), (26) as:

$$\boldsymbol{T} - \kappa\bar{T}\Big(\mathbf{W}^1\boldsymbol{P} + \mathbf{W}^2\frac{1+\epsilon}{2}\Delta t\boldsymbol{S}\Big) = \boldsymbol{R}_T, \tag{C1}$$

$$(1 - B_{1/2})\boldsymbol{P} = -\mathbf{C}\mathbf{W}^3\frac{1+\epsilon}{2}\Delta t\boldsymbol{D} + \mathbf{C}\boldsymbol{R}_P, \tag{C2}$$

$$\frac{1+\epsilon}{2}\Delta t\boldsymbol{S} = \mathbf{I}\Big[-\mathbf{B}\boldsymbol{P} - \tilde{\mathbf{C}}\mathbf{W}\frac{1+\epsilon}{2}\Delta t\boldsymbol{D} + \tilde{\mathbf{C}}\boldsymbol{R}_P\Big], \tag{C3}$$

where $\kappa = R_d/c_{pd}$, $\mathbf{C}$ is the matrix of mid-point rule integration from the model top to the ground, $\mathbf{C}_{k,l} = \Delta\eta_l$, $k,l \in [1,\mathrm{Nlev}]$, $\mathbf{W}^m$, $m \in [1,3]$ are diagonal matrices, $W_{l,l}^1 = \frac{B_l\bar{p}_{\rm s}}{A_l p_0 + B_l\bar{p}_{\rm s}}$, $W_{l,l}^2 = \frac{\bar{p}_{\rm s}}{A_l p_0 + B_l\bar{p}_{\rm s}}$, $W_{l,l}^3 = \frac{\Delta A_l p_0 + \Delta B_l\bar{p}_{\rm s}}{\bar{p}_{\rm s}\Delta\eta_l}$, $\mathbf{I}$ is the two-diagonal matrix interpolating $\dot{s}^{n+1}$ to the integer vertical levels, $\mathbf{I}_{l,l} = \mathbf{I}_{l,l-1} = \frac{1}{2}$. Matrix $\mathbf{B}$ is diagonal, $\mathbf{B}_{l,l} = (B_{l+1/2} - B_{l-1/2})$. Matrix $\tilde{\mathbf{C}}$ equals to the lower triangle (including main diagonal) of $\mathbf{C}$ and represents the operation of mid-point rule integration from model top to some level $l + 1/2$.

The definition of $G$ (21) in the column-matrix form is

$$\boldsymbol{G} = \boldsymbol{\Phi}_{\rm s} + \mathbf{A}\boldsymbol{T} + R_d\bar{T}\boldsymbol{P}, \tag{C4}$$

with the matrix of vertical integration using trapezoidal rule $\mathbf{A} = \mathbf{U}\boldsymbol{\Phi}$. $\boldsymbol{\Phi}$ is two-diagonal matrix which represents the increment for linear part of the geopotential between the neighbouring levels: $\boldsymbol{\Phi}_{\mathrm{Nlev},\mathrm{Nlev}} = \ln\frac{\bar{p}_{\rm s}}{\bar{p}_{\mathrm{Nlev}}}$, $\boldsymbol{\Phi}_{l,l} = \boldsymbol{\Phi}_{l,l+1} = \frac{1}{2}\ln\frac{\bar{p}_l+1}{\bar{p}_l}$, $l \in [1,\mathrm{Nlev}-1]$, $\mathbf{U}$ is the upper triangular matrix responsible for summation of the increments, $U_{l,k} = 1$, $l \geq k$, $l \in [1,\mathrm{Nlev}]$.

One can see that elimination of $\boldsymbol{T}$, $\boldsymbol{P}$, $\boldsymbol{S}$ from Eq. (C4) using Eqs. (C1)-(C3) results in Eq. (27) with $\mathbf{M}' = \kappa R_d\bar{T}(1-B_{1/2})^{-1}\mathbf{A}[\mathbf{W}^1\mathbf{C} + \mathbf{W}^2\mathbf{I}(\tilde{\mathbf{C}} - \mathbf{B}\mathbf{C})] + R_d\bar{T}(1-B_{1/2})^{-1}\mathbf{C}$, $\mathbf{M} = \mathbf{M}'\mathbf{W}^3$.

*Author contributions.* All the authors contributed to SL-AV20 development. Mikhail Tolstykh is the principal developer of the previous version of SL-AV model, Vladimir Shashkin developed IMC version of the model, Rostislav Fadeev constructed the algorithm for variable resolution reduced grid generation and contributed to the infrastructure of SL-AV20 model, Gordey Goyman developed iterative time-stepping scheme and carried out the numerical experiments. Vladimir Shashkin wrote first-guess article text, all co-authors contributed to model validation and the final form of article.

*Acknowledgements.* The research has been carried out at the Institute of Numerical Mathematics of the Russian Academy of Sciences and funded by the Russian Science Foundation (Grant No.14-27-00126). Authors thank Sergey Kostrykin (INM RAS) for proof-reading the paper.



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



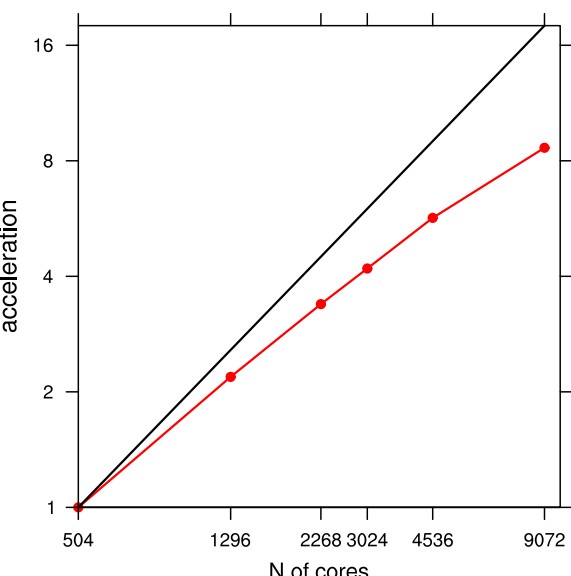

**Figure 1.** Parallel acceleration of SL-AV20 experimental configuration on the grid of 3024 by 1513 points with respect to 504 cores (red line), linear acceleration - black line.





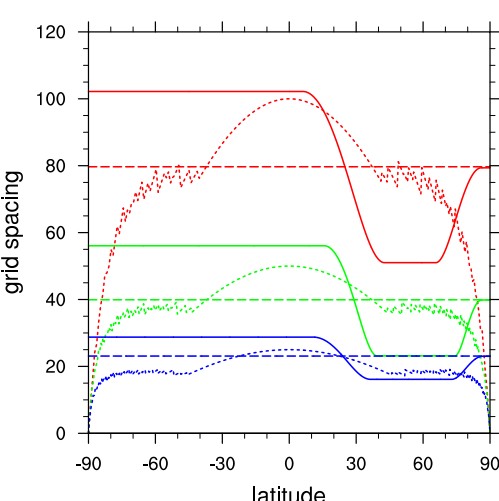

**Figure 2.** Latitude grid spacing (solid line) of $400 \times 250$ (red), $800 \times 500$ (green) and $1600 \times 865$ (blue) variable resolution latitude-longitude grids. Dotted lines presets longitude grid spacing of the corresponding reduced grids. Dashed lines shows latitude grid spacing in the constant resolution grids with the same number of latitudes.




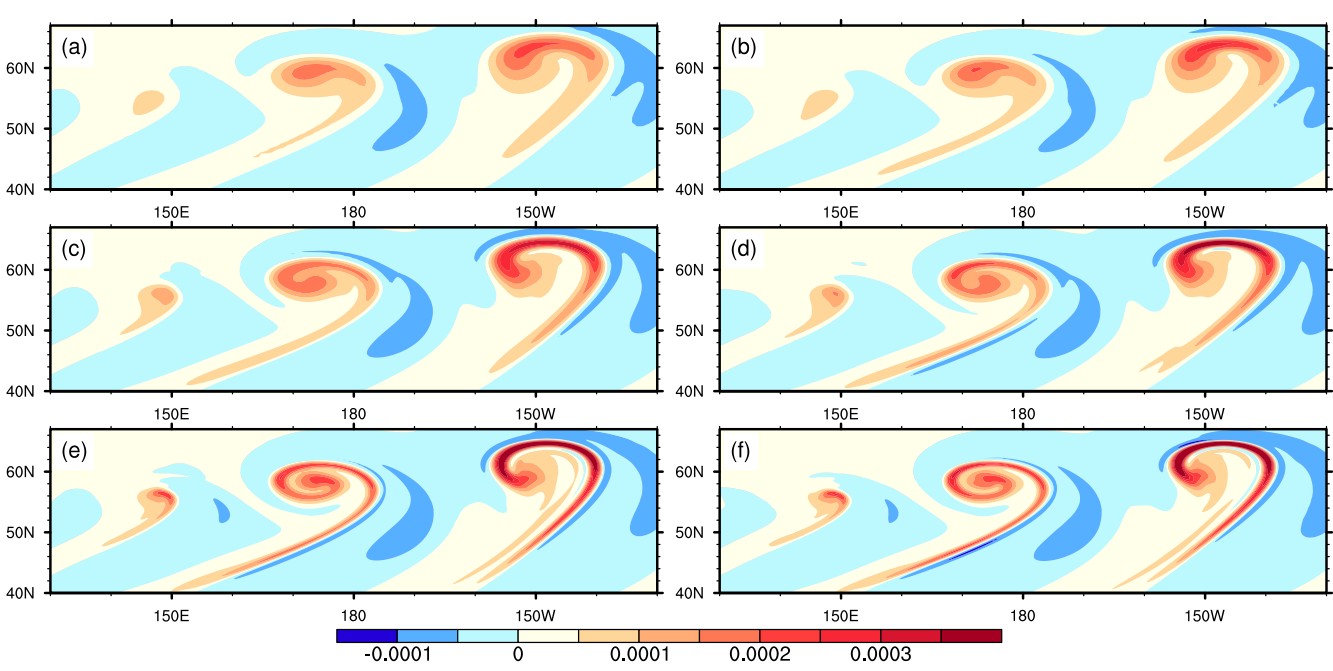

**Figure 3.** Day 9 relative vorticity at 850 hPa in baroclinic instability test. Left column – SL-AV20 solutions using regular lat-lon grids with constant latitude resolution , (a) $400 \times 250$, (c) $800 \times 500$, (d) $1600 \times 865$. Right column, (b), (d), (e) - the same as left column, but using reduced lat-lon grids with variable resolution in latitude.





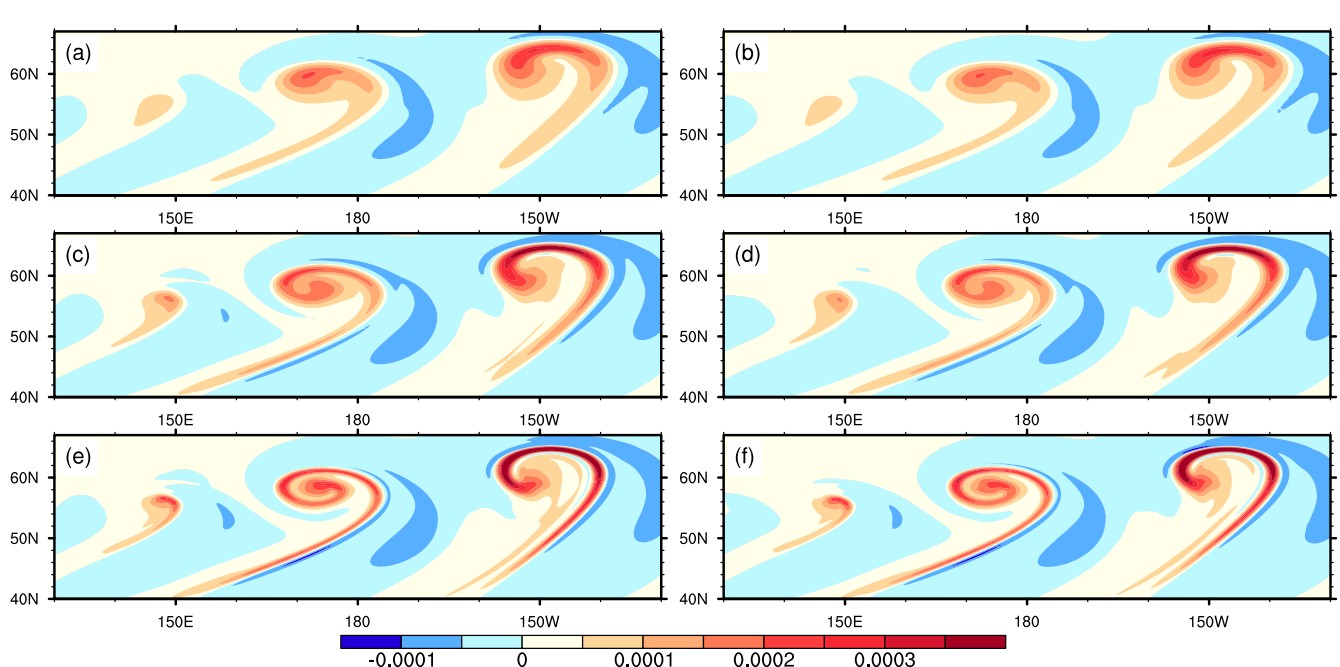

**Figure 4.** The same as Fig. 3, but lat-lon grids with variable resolution in latitude (no reduction) in left column.





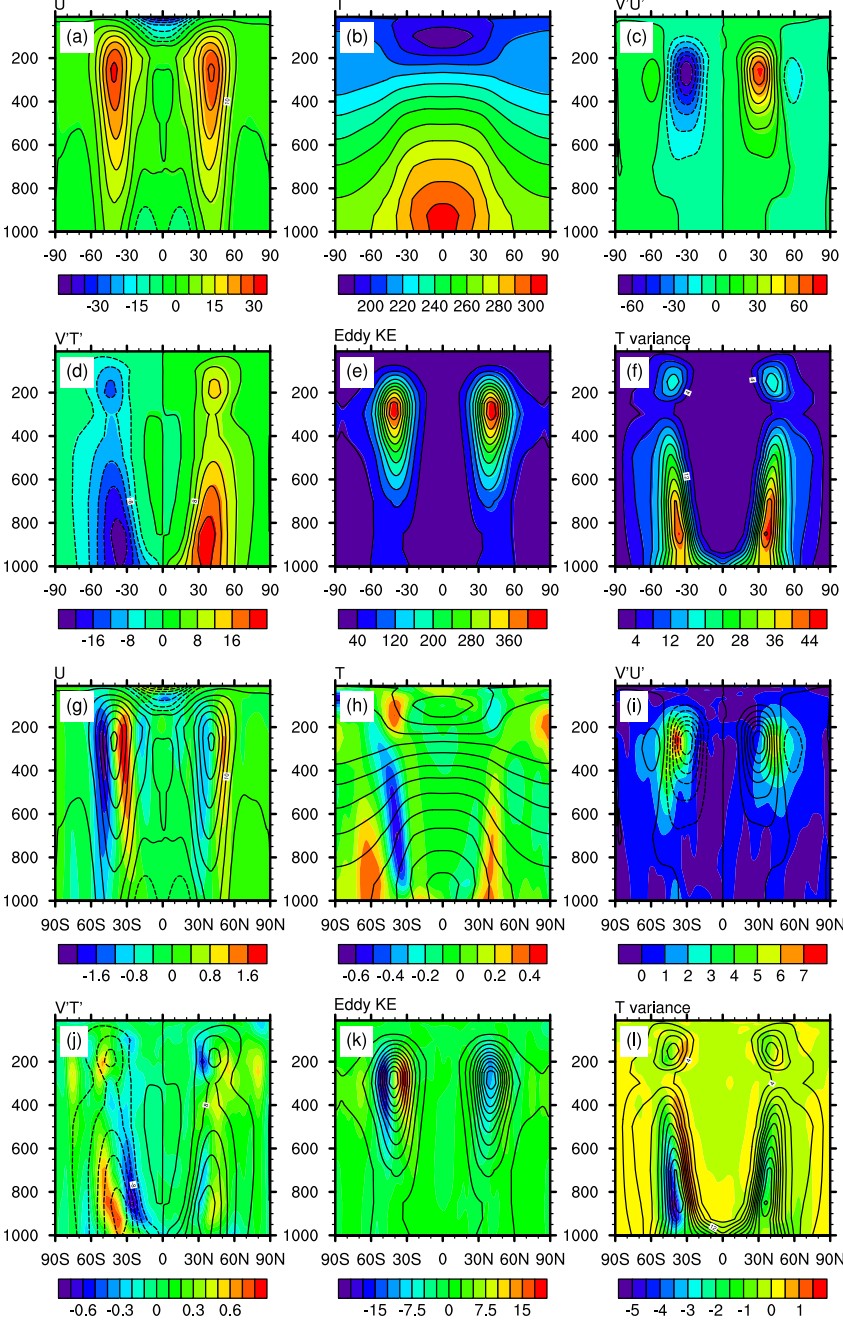

**Figure 5.** 1000 days time-zonal averaged zonal wind speed (U), temperature (T), momentum flux (U'V'), heat flux (V'T'), eddy kinetic energy and temperature variance in Held-Suarez test. Panels (a) – (f) – shading shows reduced grid solution with variable resolution in latitude, contours show regular grid solution with constant latitudinal resolution. Panels (g) – (l) – shading shows the difference between regular and reduced grid solutions with constant resolution in latitude (regular minus reduced), overlaid contours depict the regular grid solution.





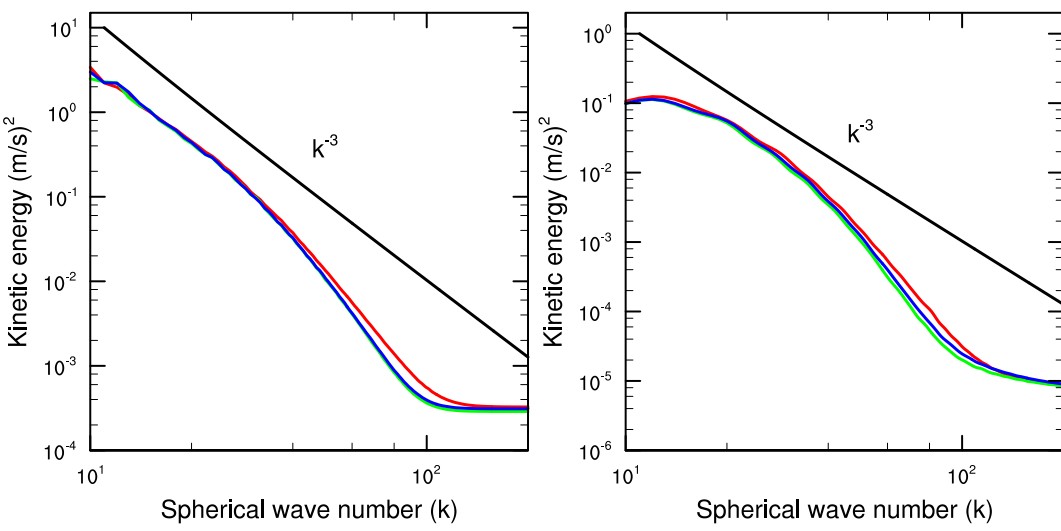

**Figure 6.** 500 hPa kinetic energy spectra in mid- and high- latitudes in Held-Suarez test. Reduced grid with variable resolution in latitude (red line), reduced grid with constant latitudinal resolution (green line), regular grid with constant latitudinal resolution (blue line).