# Peer review of "Vorticity-Divergence semi-Lagrangian Global Atmospheric Model SL-AV20: Dynamical Core"

_Geoscientific Model Development, 2016_

## Referee Comment (RC1) · Anonymous Referee #1 · 23 Dec 2016

General Comments:

This manuscript deals with the development of a hydrostatic global atmospheric dynamical core based on the semi-Lagrangian (SL) method with a vorticity-divergence formulation. This model combines several nice features such as the reduced latitude-longitude grid system, variable resolution option, semi-implicit time stepping and the use of conservative SL approach based on an efficient algorithm. The research involved is interesting and worth publishing in GMD. The technical aspect of the model is well described in the manuscript. However, the numerical experiment section requires improvements, and this may be addressed by performing more challenging tests for validation.

Major Comments:

(1) Sections 2, 5: The reduced latitude-longitude grid system can alleviate some major issues with the "pole problems." However, it is not clear whether your reduced grid system includes the singular pole points in the computations. The spherical operators in Eqn.2-5, involves terms with the cosine of the latitude in the denominator, which may lead to instability at the vicinity of the poles when using FD discretization. How do you address this issue? Please provide some description on this in the revised manuscript.

(2) Section 3.2: The SL computational efficiency is obtained with the dimension-splitting conservative cascade scheme (CCS). This scheme uses an efficient sequence of 1D operations for multi-dimensional problems. Authors should briefly outline the CCS for the sake of clarity, which would help the readers. Moreover, the basic paper on CCS algorithm (Nair et al. (2002), MWR, vol.130, pp 2059-2073) should be cited in the revision.

(3) Section 9: The performance of the model is validated with a couple of experiments. The J-W baroclinic instability test is a relatively simple test for SL models, both reduced and full grid models produce very similar results. The Held-Suarez test shows the time-space averaged results over 1000 days, it is not a challenging test for comparing the numerical schemes or grid systems. What it shows is the model's overall ability to maintain an equilibrium for long-term integrations.

Authors should consider performing a short-term integration experiment, based on flow over an isolated mountain, similar to the SW test-case 5 proposed by Williamson et al. (1992). The mountains/topography pose problems particularly for the SL models, and such a test would be far more interesting. See the "mountain-induced Rossby wave test" in Simmaro et al. (2013), Tellus A 2013, 65, 20270, http://dx.doi.org/10.3402/tellusa.v65i0.20270).

Also the reference: Jablonowski, C., Lauritzen, P. H., Taylor, M. A. and Nair, R. D. 2008. Idealized test cases for the dynamical cores of atmospheric general circulation models:

a proposal for the NCAR ASP 2008 summer colloquium. http://esse.engin.umich.edu/admg/publications.php

Minor Comments:

(1) In the Abstract please indicate that your model is hydrostatic, these days dynamical cores mostly imply non-hydrostatic model development. (2) Reference: On Page 26, lines 25-28: The two references for baroclinic instability tests by Jablonowski, C. and Williamson, D... refer to same test, keep any one of them. (3) It would be nice to include your future extensions plans (if any) with this hydrostatic dynamical core, in the Conclusions.

---

## Referee Comment (RC2) · Anonymous Referee #2 · 9 Jan 2017

[english]article [a4paper]geometry verbose,tmargin=2cm,bmargin=3cm,lmargin=3cm,rmargin=3cm babel

**Review on paper: "Vorticity-Divergence semi-Lagrangian Global Atmospheric Model SL-AV20:Dynamical Core"**

January 9, 2017

**1   Overview**

This paper describes the new version of the hydrostatic SL-AV20 hydrostatic dynamical core used by the Russian Hydrometeorological Centre for global weather forecasting. It is based on a vorticity-divergence formulation and semi-Lagrangian semi-implicit timestepping using a reduced lat/lon unstaggered grid to reduce computational cost. The model formulation and the numerical discretizations are described in the paper and the reduced grid version of the model is compared against a standard (unreduced) lat/lon grid version on two well known test cases for dynamical cores.

SL-AV20 dynamical core has noticeable differences from other hydrostatic semi-implicit, semi-Lagrangian dynamical cores both in the formulation and the implementation. Some of these aspects are novel and therefore I would be happy to recommend publication of this paper in GMD, however, it requires some revision as outlined in my comments in the following sections.

**2    Major comments**

Section 2 describes the governing equations. The paper relies to some degree on reader's familiarity with the particular approach followed in SL-AV. For this reason this section needs few modifications to improve its clarity. In particular:

1. Eq (1). It is mentioned that it is derived by applying $k \cdot \nabla \times$ to the momentum equations. I think that it would be better to start from (and explicitly write) the original form of momentum equations that give Eq (1). This must be Eq (3). In this case lines 5-10 need some re-arrangement so that description evolves from the simpler to the more complex form. It is not very clear where the coefficient $B(\eta)p_s/(A(\eta)p_0 + B(\eta)p_s)$ comes from and would be helpful to guide the reader on this. I wonder if the Rochas, 1990 reference is available through a web-link? If that is so please provide it.

2. Please add some information on how you derived Eq (6). I assume that this equation is the analogue of Eq (15) of McDonald and Haugen 1993 if you consider $T_v$?

3. Continuity equation (8). Please explain how is this derived from original Eq (2) of McDonald and Haugen 1993 or any other form you may have considered.

In the topics of implementation and numerical experiments I have the following questions/comments:

1. Please comment somewhere in the paper on the efficiency/scalability of their proposed approach compared with other similar hydrostatic spectral semi-Lagrangian approaches on reduced Gaussian grids where transpositions are also necessary for the Fourier/Legendre transforms.

2. For the experiments in section 9, the timestep used is 1200s reduced by a resolution depending factor c as the resolution increases. What is the maximum CFL

in these experiments? Given that SISL timestepping is used the model should be able to run stably at max CFL larger than 1 (e.g. 5) without loss of accuracy.

3. Is there a standard verification comparison (e.g. 500hPa geopotential height RMSE / Anomaly Correlation Coefficient) between fixed and reduced SL-AV20 model on real forecast cases available? If such comparison exists I would recommend to include it to strengthen the validation part of the paper.

**3 Minor comments**

I think that the fact that this is a hydrostatic dynamical core should appear very early in the text and in the abstract.

1. Line 8: "test cases"

2. page 2, lines 4-5. I think that for "computational efficiency" we want to achieve a solution at a given accuracy at the shortest possible time for a given number of processors.

3. page 2, line 8: perhaps "few kilometers" is meant instead of "first kilometers"?

4. page 2, line 16: "cost" instead of "pattern"?

5. page 21, line 9: "deterministic"

---

## Author Comment (AC1) · 7 Mar 2017

Dear Reviewer, thank you for finding time to read our article and present your comments that helped to improve the manuscript. Please find our responses below.

**Author responses to Reviewer 1 comments**

**Major Comments:**

(1) Sections 2, 5: The reduced latitude-longitude grid system can alleviate some major issues with the "pole problems." However, it is not clear whether your reduced grid system includes the singular pole points in the computations. The spherical operators in Eqn.2-5, involves terms with the cosine of the latitude in the denominator, which may lead to instability at the vicinity of the poles when using FD discretization. How do you address this issue? Please provide some description on this in the revised manuscript.

Answer: Our reduced grid does include the pole points and this was indicated in Sect. 5.1. We recognize two problems with $cos\varphi$ denominator: (1) how to approximate the operators at pole points, (2) the denominator is small in the vicinity of poles that can lead to instability. Our approach to the first problem was completely explained in Sect. 5.2. Additionally, we corrected a mistake in the denominator of Eq. 36 and added Eqs. 37, 38 that extend 36 to the pole points. As for the second problem, we believe that it is really dangerous only if short zonal waves are present in the polar regions. The reduced grid does help to avoid the instability, because it does not support the most of short zonal waves in the polar regions. On the full grid, fortunately, the $\nabla^4$ diffusion with the implicit time-stepping effectively dumps short zonal waves in the vicinity of poles (as shown by ref. Li & Bates 1994). The following sentence is added in Sect. 7.1 ("Fourth order hyper-diffusion") in the paragraph after Eq. (56). So we never noticed any instability that we can attribute to $cos\varphi$ denominator.

(2) Section 3.2: The SL computational efficiency is obtained with the dimension-splitting conservative cascade scheme (CCS). This scheme uses an efficient sequence of 1D operations for multi-dimensional problems. Authors should briefly outline the CCS for the sake of clarity, which would help the readers. Moreover, the basic paper on CCS algorithm (Nair et al. (2002), MWR, vol.130, pp 2059-2073) should be cited in the revision.

Answer: The requested information is included at the end of Sect. 3.2.

(3) Section 9: The performance of the model is validated with a couple of experiments. The J-W baroclinic instability test is a relatively simple test for SL models, both reduced and full grid models produce very similar results. The Held-Suarez test shows the time-space averaged results over 1000 days, it is not a challenging test for comparing the numerical schemes or grid systems. What it shows is the model's overall ability to maintain an equilibrium for long-term integrations. Authors should consider performing a short-term integration experiment, based on flow over an isolated mountain, similar to the SW test-case 5 proposed by Williamson et al. (1992). The mountains/topography pose problems particularly for the SL models, and such a

test would be far more interesting. See the "mountain-induced Rossby wave test" in Simmaro et al. (2013), Tellus A 2013, 65, 20270, http://dx.doi.org/10.3402/tellusa.v65i0.20270). Also the reference: Jablonowski, C., Lauritzen, P. H., Taylor, M. A. and Nair, R. D. 2008. Idealized test cases for the dynamical cores of atmospheric general circulation models: a proposal for the NCAR ASP 2008 summer colloquium. http://esse.engin.umich.edu/admg/publications.php

Answer: The results of mountain induced Rossby wave test case are now included in the revised manuscript, please see Sect. 9.3.

**Minor Comments:**

(1) In the Abstract please indicate that your model is hydrostatic, these days dynamical cores mostly imply non-hydrostatic model development. (2) Reference: On Page 26, lines 25-28: The two references for baroclinic instability tests by Jablonowski, C. and Williamson, D. . . refer to same test, keep any one of them. (3) It would be nice to include your future extensions plans (if any) with this hydrostatic dynamical core, in the Conclusions.

Answer: All comments are accepted, the corresponding changes are made, except (2): we decided to keep both references since JW2006b is the "main" reference for this test case in high-impact journal and JW2006a contains some information (pictures for high resolution reference solutions) not present in JW2006b.

---

## Author Comment (AC2) · 7 Mar 2017

Dear Reviewer, thank you for finding time to read our article and present your comments that helped to improve the manuscript. Please find our responses below.

**Author responses to Reviewer 2 comments**

**Major Comments:**

Section 2 describes the governing equations. The paper relies to some degree on reader's familiarity with the particular approach followed in SL-AV. For this reason this section needs few modifications to improve its clarity. In particular:

1. Eq (1). It is mentioned that it is derived by applying $k \cdot \nabla \times$ to the momentum equations. I think that it would be better to start from (and explicitly write) the original form of momentum equations that give Eq (1). This must be Eq (3). In this case lines 5-10 need some re-arrangement so that description evolves from the simpler to the more complex form. It is not very clear where the coefficient $B(\eta)p_s / (A(\eta)p_0 + B(\eta)p_s)$ comes from and would be helpful to guide the reader on this. I wonder if the Rochas, 1990 reference is available through a web-link? If that is so please provide it.

Answer: We introduced the requested changes in Sect. 1 (see Eq. 1 and disscussion next to it in the revised manuscript). Unfortunately, we cannot find ref. Rochas, 1990 through the web. However, it seems that Temperton, 1997 repeated the principal information from Rochas, 1990, so we include this link in the revised manuscript.

2. Please add some information on how you derived Eq (6). I assume that this equation is the analogue of Eq (15) of McDonald and Haugen 1993 if you consider $T_v$?

Answer: The information on derivation of thermodynamic equation for $T_v$ is added in the revised manuscript (please see discussion before Eq. 6).

3. Continuity equation (8). Please explain how is this derived from original Eq (2) of McDonald and Haugen 1993 or any other form you may have considered.

Answer: Done, please see Eqs. 8,9 of the revised manuscript and theirs discussion.

In the topics of implementation and numerical experiments I have the following questions/comments:

1. Please comment somewhere in the paper on the efficiency/scalability of their proposed approach compared with other similar hydrostatic spectral semi-Lagrangian approaches on reduced Gaussian grids where transpositions are also necessary for the Fourier/Legendre transforms.

Answer: The following comment is added at the end of Sect. 8 of revised manuscript.

2. For the experiments in section 9, the timestep used is 1200s reduced by a resolution depending factor c as the resolution increases. What is the maximum CFL in these experiments? Given that SISL timestepping is used the model should be able to run stably at max CFL larger than 1 (e.g. 5) without loss of accuracy.

Answer: We followed the recommendation of (Jablonowski and Williamson 2006) to test the model in its operational configuration. However, the time-step values consistent with operational practice in the presented idealized test lead to only moderate advective CFL numbers, so we repeated the experiments with time-steps up to 3 times larger and obtained very similar results. The information on the large-time step experiments is included in the revised version of manuscript (Sect. 9.1, Sect. 9.2). Nevertheless, it should be mentioned that the magnitude of non-linear terms of discretized equations is the real stability challenge for SISL models, not the advective CFL. We believe that the magnitude of non-linear terms in J&W2006 test is similar to typical values for real troposphere, thus it is very hard to be stable in this experiment with time-step providing really amazing advective CFL of > 5.

3. Is there a standard verification comparison (e.g. 500hPa geopotential height RMSE / Anomaly Correlation Coefficient) between fixed and reduced SL-AV20 model on real forecast cases available? If such comparison exists I would recommend to include it to strengthen the validation part of the paper.

Answer: The current operational version of SL-AV20 does not use reduced grid. Unfortunately, there is no computer resources currently available to make necessary parallel runs for complete testing and tuning of reduced grid version (current HMCR resources are limited just to 50 Tflops peak) . So, we'd better not to build up on the individual forecasts results and leave the detailed comparison of regular and reduced grid configurations performance in real-flow forecasting to be the matter of future work . Anyway, next SLAV version will not be able to run properly without reduced grid.

**Minor comments:**

I think that the fact that this is a hydrostatic dynamical core should appear very early in the text and in the abstract.
1. Line 8: "test cases"
2. page 2, lines 4-5. I think that for "computational efficiency" we want to achieve a solution at a given accuracy at the shortest possible time for a given number of processors.
3. page 2, line 8: perhaps "few kilometers" is meant instead of "first kilometers"?
4. page 2, line 16: "cost" instead of "pattern"?
5. page 21, line 9: "deterministic"

Answer:  All points are corrected, thank you very much.

---

## Author Comment (AC3) · 7 Mar 2017

Comment to the Topical Editor on the code availability

Unfortunately, currently we are not allowed to place SL-AV20 source code at some public service (like git-hub etc.). However, the code is available (and will be available in the future) through the e-mail contact to Mikhail Tolstykh as indicated in the code availabilty section, so every interested reader can get it.

---

## Author Response (AR2)

Dear Dr. James R. Maddison, thank you very much for treating our paper as a topical editor. Please find author's responses to your comments below.

1. The GMD code policy indicates that, if code cannot be publicly released, that reasons for this be provided. Please can you add this to the "Code Availability" section of the article.

The reasons why the code is available only through e-mail request to Mikhail Tolstykh are described now in the "Code availability" section. 'Certain limitations' in the text mean that SL-AV20 is a fruit of collaboration between INM RAS and Hydrometcentre (belongs to Roshydromet, Russian meteoservice), and we would like to avoid touching Roshydromet bureaucracy about intellectual property.

2. I think it would be helpful to add further details regarding previous developments, mentioned at the top of page 2. In particular a summary of the different aspects, developed or described in the supplied references, could be supplied here.

We agree, the short summary of work done in the supplied references is added at the top of p. 2.

3. The comment regarding performance, starting at the end of line 25 on page 18, seems somewhat speculative without supporting performance comparisons.

We have decided to omit this sentence.

4. The sentences starting on lines 10 and 14 of page 11, line 17 of page 17, lines 12-13 of page 18, lines 1 and 9 of page 19, the end of line 16 of page 20, and the sentence on line 28 of page 22, are a bit informal and should be rephrased.

The mentioned sentences are corrected.

5. It would be helpful to define parallel efficiency on page 18.

We added the definition, it is really helpful, thank you!

We have also introduced some minor language corrections.

[revised manuscript text omitted]